# Simplified internal models in human control of complex objects

**Salah Bazzi**[1]*, **Stephan Stansfield**[2], **Neville Hogan**[2,3], **Dagmar Sternad**[1,4,5,6]

1 Institute for Experiential Robotics, Northeastern University, Boston, Massachusetts, United States of America, 2 Department of Mechanical Engineering, MIT, Cambridge, Massachusetts, United States of America, 3 Department of Brain and Cognitive Sciences, MIT, Cambridge, Massachusetts, United States of America, 4 Department of Biology, Northeastern University, Boston, Massachusetts, United States of America, 5 Department of Electrical and Computer Engineering, Northeastern University, Boston, Massachusetts, United States of America, 6 Department of Physics, Northeastern University, Boston, Massachusetts, United States of America

* s.bazzi@northeastern.edu

**Data Availability Statement:** All the replication data for "Simplified Internal Models in Human Control of Complex Objects" is located in a link at the following repository: https://doi.org/10.7910/

## Abstract

Humans are skillful at manipulating objects that possess nonlinear underactuated dynamics, such as clothes or containers filled with liquids. Several studies suggested that humans implement a predictive model-based strategy to control such objects. However, these studies only considered unconstrained reaching without any object involved or, at most, linear mass-spring systems with relatively simple dynamics. It is not clear what internal model humans develop of more complex objects, and what level of granularity is represented. To answer these questions, this study examined a task where participants physically interacted with a nonlinear underactuated system mimicking a cup of sloshing coffee: a cup with a ball rolling inside. The cup and ball were simulated in a virtual environment and subjects interacted with the system via a haptic robotic interface. Participants were instructed to move the system and arrive at a target region with both cup and ball at rest, 'zeroing out' residual oscillations of the ball. This challenging task affords a solution known as 'input shaping', whereby a series of pulses moves the dynamic object to the target leaving no residual oscillations. Since the timing and amplitude of these pulses depend on the controller's internal model of the object, input shaping served as a tool to identify the subjects' internal representation of the cup-and-ball. Five simulations with different internal models were compared against the human data. Results showed that the features in the data were correctly predicted by a simple internal model that represented the cup-and-ball as a single rigid mass coupled to the hand impedance. These findings provide evidence that humans use simplified internal models along with mechanical impedance to manipulate complex objects.

## Author summary

In their daily life, humans interact with objects that possess complex dynamics. Some examples include picking up containers filled with liquids, tying shoelaces, and folding laundry. To allow for dexterous interactions with these objects, we hypothesized that the

DVN/D0GYHV. The dynamic simulation and optimization code used in this study may be found at https://github.com/stephan-stansfield/cup-task.

**Funding:** This research was supported by the National Institutes of Health (www.nih.gov) (R37-HD087089 and R01-CRCNS-NS120579 to DS), the Eric P. and Evelyn E. Newman Fund (website N/A) (to NH), and the National Science Foundation Graduate Research Fellowship Program (www.nsf.gov) (1745302 and 2141064 to SS). Any opinions, findings, and conclusions or recommendations expressed in this material are those of the authors and do not necessarily reflect the views of the National Science Foundation. The funders did not play any role in the study design, data collection and analysis, decision to publish, nor preparation of the manuscript.

**Competing interests:** The authors have declared that no competing interests exist.

brain develops an internal representation of their dynamics. But what form does this representation take? Is it a detailed representation or an approximate one? To answer these questions, we had subjects interact with a simplified cup of coffee in a virtual environment: a 2D cup moving on a horizontal line, and with a ball rolling inside. Subjects were instructed to move the cup from a start position and quickly arrive at a target position, with both the cup and ball completely stopped, i.e., the ball should not be rolling once the cup reached the target position. The manner by which subjects moved the cup-and-ball prior to arriving at the target location allowed us to infer what internal representation subjects developed of the system's dynamics. Experimental and modeling results showed that subjects developed a simple and approximate representation of the system's dynamics rather than a complex and detailed one.

## Introduction

Humans manipulate and interact with a myriad of objects on a daily basis. From rigid objects like pens and hammers to non-rigid ones like clothes and containers filled with liquid, the dynamics governing the behavior of these objects vary in complexity. Most studies on object manipulation in motor neuroscience have focused on simple tasks involving rigid objects: grasping [1], lifting [2], and transporting [3]. In these tasks, controlling the movement of the hand is equivalent to controlling the movement of the object, only with an added mass. Relatively few studies have examined human control of objects that contain internal degrees of freedom, i.e., non-rigid objects. Non-rigid objects are underactuated, meaning their internal degrees of freedom are not controlled directly, but rather indirectly through the interaction between the hand and the object. Manipulation of such objects is considerably more challenging than unconstrained reaching or rigid object manipulation. It is yet to be fully understood how humans manage to interact with objects with such complex dynamics in their numerous daily activities.

The few existing studies on non-rigid object manipulation modeled the human behavior as being optimal and sought to identify what criteria the humans were optimizing. All these models proposed that humans sought to maximize some measure of smoothness [4,5,6], effort [7], or both [8]. While informative, these models are confined to the descriptive level of analysis: they describe *what* the biological system does, but not *how* it is achieved. One assumption inherent to these approaches is that humans form precise internal models that specify the forces that need to be exerted on the object to induce a desired motion [7,9]. This is plausible since the non-rigid object considered in these studies was merely a linear mass-spring system, a system that does not possess the considerably more complex nonlinear dynamics of the broad range of objects that humans interact with daily. Nonlinear underactuated systems are very sensitive to small changes in their input and initial conditions, and some are even prone to chaos [10]. Such small perturbations readily arise from the fact that human actions are intrinsically variable. Can humans acquire precise internal models of more complex objects with nonlinear and potentially chaotic dynamics? What level of detail do these internal models represent?

Feedback can handle imprecision and can alleviate the need for an accurate internal model. One study showed that optimal feedback control can explain the behavior observed in human control of non-rigid objects, specifically mass-spring systems [7]. However, for more complex objects the behavior can evolve on a very short time scale and become unpredictable, rendering feedback inadequate due to the long delays in neural transmission. Alternatively, the

spring-like properties of muscles, specifically mechanical impedance, may be used to handle discrepancies between predicted and actual movement [8,11]. How do humans exploit muscle impedance in conjunction with internal models to control complex objects?

To develop a deeper understanding of dexterous sensorimotor control more challenging tasks are required [12,13]. In addition to interacting with a complex object, namely a container with a rolling ball inside, this study increased the task's demands with a challenging goal: move the system, i.e., the cup and also the ball, to come to a complete rest. This instruction of 'zeroing out' terminal oscillations of the ball presented a major challenge for the human subjects, especially as they had to move very fast. Participants were instructed to transport this simplified 'cup of coffee' in a virtual environment. This system was presented to the subjects as a cup with one internal degree of freedom, a ball, rolling inside the cup. The physics that were virtually simulated were those of a cart-pendulum system, which was already employed in our previous research [10,14,15,16,8].

For this task of zeroing out residual oscillations, control engineering has developed a strategy named 'input shaping' that achieves a perfect solution. However, knowledge of the system being controlled is necessary. Input shaping applies a series of timed pulses to move a dynamic object from point to point leaving no residual oscillations [17]. Since the timing and amplitude of these pulses depend on the controller's internal model of the object, input shaping could serve as a tool to estimate the human's internal representation of the cup-and-ball and, hence, probe human control. Simulations employing five internal models with different degrees of complexity were compared against the human data. The internal models ranged from a detailed representation of the coupled hand and cup-and-ball system to a rigid body model without the oscillatory dynamics of the ball. Remarkably, the latter model best reproduced the data.

## Results

### Experimental task

The task of transporting a 'cup of coffee' was simplified to moving a two-dimensional arc with a ball 'rolling' inside (Fig 1). This cup-and-ball system was modeled as a cart-and-pendulum system, where the cup's movement corresponded to the cart's movement and the ball represented the pendulum's bob (Fig 1A). Even though this model simplified the full dynamics of a cup of coffee, the essential challenges of the task were retained: underactuation and nonlinearity, where the latter could induce chaotic behavior. The motion of the cup was limited to the horizontal axis only (Fig 1B). The equations of motion for this system were:

$$(m_c + m_p)\ddot{x} = m_p l(\dot{\phi}^2 \sin \phi - \ddot{\phi} \cos \phi) + F = F_{ball} + F \tag{1}$$

$$\ddot{\phi} = -\frac{\ddot{x}}{l} \cos \phi - \frac{g}{l} \sin \phi \tag{2}$$

where $x$ denoted the cup position and $\phi$ denoted the ball angle, with $\phi = 0$ defined at the bottom of the cup; positive angles moved counterclockwise. The mass of the cup was $m_c$ and the pendulum's mass and length were $m_p$ and $l$, respectively. $F$ was the force applied by the subject on the cup and $g$ was the gravitational acceleration. $F_{ball}$ denoted the force that the ball applies back onto the cup. The cup-and-ball system was simulated in a virtual environment and visually displayed on a projection screen. Throughout the experiment, the following values were used: $m_c = 1.9$ kg, $m_p = 1.1$ kg, $l = 0.5$ m, and $g = 9.81$ m/s$^2$. Subjects interacted with the cup-and-ball using the HapticMaster robotic manipulandum [18].

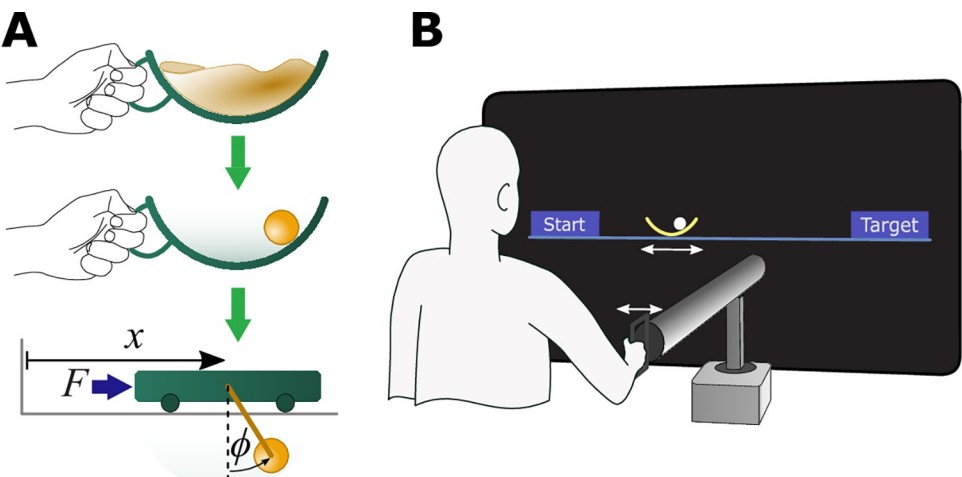

**Fig 1. Experimental task and apparatus. A.** Inspired by the daily task of carrying a cup of coffee, a simplified model of a cup with a ball rolling inside was designed. Mechanically, this cup-and-ball system was equivalent to a cart-and-pendulum system. **B.** Subjects interacted with the cart-and-pendulum system in a virtual environment through a robotic manipulandum. The cart-and-pendulum system was visually displayed as a 2D semicircular cup with a ball (the pendulum's bob) sliding inside. The robotic manipulandum provided haptic feedback about the forces that the ball imparted on the cup.

At the beginning of each trial, the cup was centered inside the start region on the left of the screen with the ball resting at the bottom of the cup. Subjects made one-dimensional movements from the start region to the target region, and were instructed to arrive at the target region with no residual ball oscillations, i.e., both the cup and the ball should come to a complete stop. As the task was difficult, subjects were trained in blocks 1 and 2 before performing the desired task in blocks 3 and 4.

In the first block of 25 trials, subjects were instructed to move the cup-and-ball system from the start region and bring the cup to a complete stop inside the target region within 1.50 s. Subjects did not have to pay attention to the movements of the ball, except for making sure they did not lose the ball. For the second block of trials, subjects practiced moving the cup-and-ball to arrive at the target region with no residual ball oscillations. Subjects' performance was quantified using the *residual ball angle*, which was defined as the maximum absolute ball angle after the cup had arrived at the target region. Subjects were verbally encouraged to try to move at a pace similar to the first block. After the first two familiarization blocks, subjects were presented with the two testing blocks 3 and 4, where they were required to both minimize residual ball oscillations and meet a specific movement duration constraint: 1.40 s in block 3 and 1.20 s in block 4. To motivate subjects to try their best, they were told that a residual angle less than or equal to 10 deg was considered skillful performance.

## Input shaping

Input shaping is a control strategy developed to move objects with resonant internal dynamics from point to point with no residual oscillation of the internal degrees of freedom at the target [17]. This is achieved by commanding impulses of appropriate amplitude and timing to the system that result in destructive interference of the excited oscillatory modes. Impulses with these amplitudes and timings are convolved with the nominal input, i.e., the input that would result in the desired displacement of the object. This produces a *shaped input* that will both command the system to the desired position and eliminate residual oscillations (Fig 2). For a

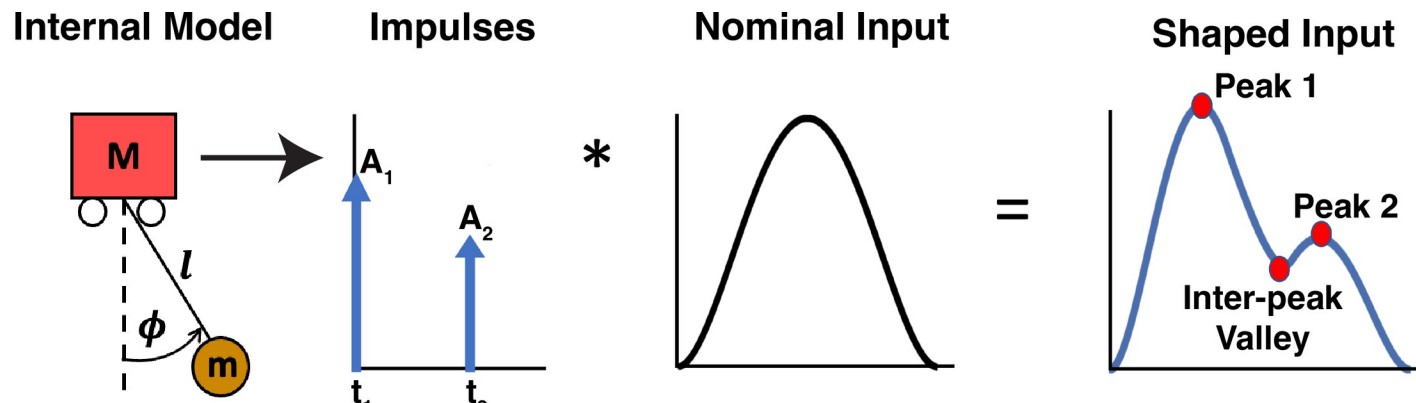

**Fig 2. Input shaping technique for generating commands that minimize residual oscillations.** First, using the internal model of the system, impulses of appropriate amplitude and timing are computed (Eq 3). These impulses are then convolved with a nominal input that would nominally result in the desired displacement of the system. The result is a shaped input that will both command the system to the desired position and eliminate residual oscillations. The peaks and valleys in the shaped input are the result of the computed impulses, which in turn represent the controller's knowledge of the system's dynamics. They can therefore be used to infer the subject's internal model of the system.

given mode of oscillation of a system, the amplitudes ($A_i$) and timings ($t_i$) of the required impulses can be computed as

$$A_1 = \frac{1}{1+C}; \; A_2 = \frac{C}{1+C}; \; t_1 = 0; \; t_2 = \frac{T_d}{2}; \; C = e^{\frac{-\zeta\pi}{\sqrt{1-\zeta^2}}} \tag{3}$$

where $\zeta$ is the damping ratio and $T_d$ is the damped natural period. In this study, the nominal input was taken to be a minimum-jerk velocity profile [19]. As can be seen in Fig 2, the shaped input consists of peaks and valleys, the amplitudes and timings of which depend on the impulses, which themselves depend on the system's dynamics. Two controllers with different information about the system's dynamics, i.e., different internal models of the system, would provide different shaped inputs to the system and result in different outputs. Therefore, the peaks and valleys observed in the subjects' behavioral data could be used to infer their internal model of the system's dynamics.

Input shaping is generally based on the system's natural frequency and damping ratio, which are parameters that are poorly defined for nonlinear systems. Several approaches have been proposed to implement input shaping for nonlinear systems, a topic that is beyond the scope of the paper [20]. However, most approaches follow the same procedure: 1) linearize the system to estimate the frequency and damping and hence the required impulses, and then 2) use mathematical techniques to guarantee robustness of the input shaper to modeling uncertainties. In this paper, we did not seek to design perfect input shapers for nonlinear systems, but rather we aimed to use input shaping as a tool to determine subjects' internal model of the nonlinear system. For that purpose, it sufficed to compute the impulses for input shaping simulations based on the frequency and damping ratio of the linearized system.

Consider the cart-and-pendulum system described by Eqs (1)–(2) with the parameters mentioned earlier. For the first set of simulations in this paper, the cup-and-ball system was linearized about the pendulum's downward static equilibrium, $\phi = 0, \dot{\phi} = 0$, resulting in the linearized equations of motion:

$$(m_c + m_p)\ddot{x} = -m_p l\ddot{\phi} + F \tag{4}$$

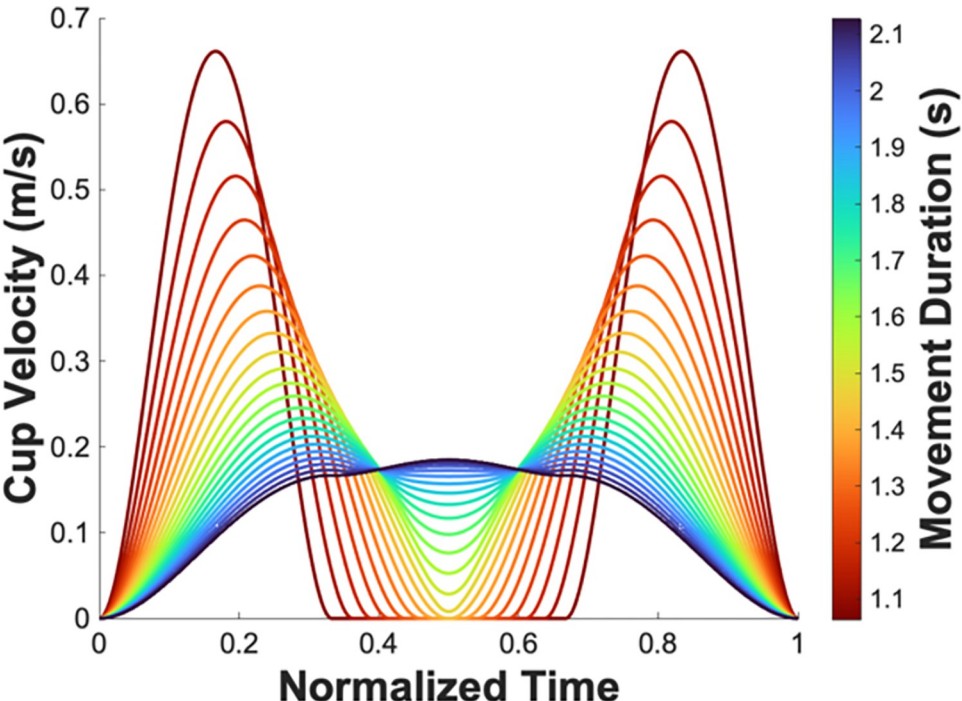

**Fig 3. Shaped inputs required to eliminate residual oscillations from the linearized cart-pendulum system (Eqs 4–5), for different movement durations.** It was assumed that the controller has a precise internal model of the system's dynamics. Two important observations are worth highlighting: 1) the equality of the two peak amplitudes for movements of sufficiently short durations, and 2) the decrease of inter-peak minimum velocities with shorter movement durations.

$$\ddot{\phi} = -\frac{g}{l}\phi - \frac{\ddot{x}}{l} \tag{5}$$

An input shaping controller provided with this precise internal model of the system's dynamics would compute the required amplitudes and timings for the impulses from (3) as $A_1 = 0.5$; $A_2 = 0.5$; $t_1 = 0$; $t_2 = 0.71$.

Fig 3 illustrates the simulated velocity profiles of shaping a minimum-jerk trajectory of 22 different durations between .75 and 1.5 times the object's natural period with these impulses [21]. A robust observation of this summary graph is the symmetry of all trajectories and the presence of two peaks of equal amplitude for any given movement of a sufficiently short duration. These two-peaked kinematic patterns are the result of the equal amplitude of the two impulses, which were computed based on the system dynamics that were provided to the controller. From Eq (3), we note that shaping the input to a system with zero damping will always result in such a symmetric two-peaked profile. Another observation is the fact that the value of the inter-peak minimum velocity was positively correlated with the movement duration: shorter movement durations had smaller minimum velocities. The inter-peak minimum velocity plateaued at 0 m/s for movement durations that were shorter than 1.42 s, which is the pendulum's natural period of oscillation.

To summarize, an input shaping controller that assumes an internal model in the form of linearized cart-and-pendulum dynamics predicts three features: 1) Two peaks in the velocity

profile, 2) equal magnitude of these two peaks, and 3) a positive correlation between the movement duration and the inter-peak minimum velocity that plateaus at around 1.42 s.

### Familiarization blocks

In block 1, the mean movement duration across all subjects was 1.36 ± 0.33 s. Most subjects (9 out of 11) achieved a mean movement duration that was shorter than the upper bound for this block (1.54 s) demonstrating that on average they learned to move the cup-and-ball system at the required pace. Fig 4A shows the average velocity profiles across all trials for each of the 11 subjects. Their movement times were normalized for better comparison of the shape. The velocity profiles displayed only one peak with an additional 'shoulder' caused by the moving ball acting upon the cup. These features were consistent across all subjects and are typical for movements that involve transporting the cup with the moving ball [22]. As to be expected, the ball kinematics displayed large residual oscillations, since the subjects did not seek to minimize these oscillations in this first block (Fig 4B).

In block 2, the aim was to familiarize the subjects with the explicit focus on minimizing residual ball oscillations while still moving at the same pace as in block 1. A first look at cup and ball kinematics in Fig 5A and 5B shows the change in cup velocity from the early to the later trials of block 2 for all subjects. At the beginning, the velocity profiles of the subjects' movements exhibited a single peak with an additional 'shoulder', similar to what is seen in block 1. However, by the end of the block all subjects had adopted a new pattern that displayed two peaks in the velocity profile. The effect of this change in movement pattern on the residual ball oscillations is shown in Fig 5C and 5D. As the subjects shifted to a two-peak velocity profile, the residual ball oscillations became slightly smaller, although there was even more variability in the patterns.

Fig 6 shows how the residual ball angles of the subjects changed as they practiced the task. Fig 6A shows ball angles over trials in two selected subjects with linear regressions: S1 shows the expected decrease with practice, while S7 shows an increase in the angle with practice, counter to the task instructions, speaking to the difficulty of the task. Linear regressions were fitted that revealed that 8 out of 11 subjects (S2-S5, S7-S10) did not show any statistically significant trends with practice (Fig 6B and Table 1). The zero slope was within the 95%

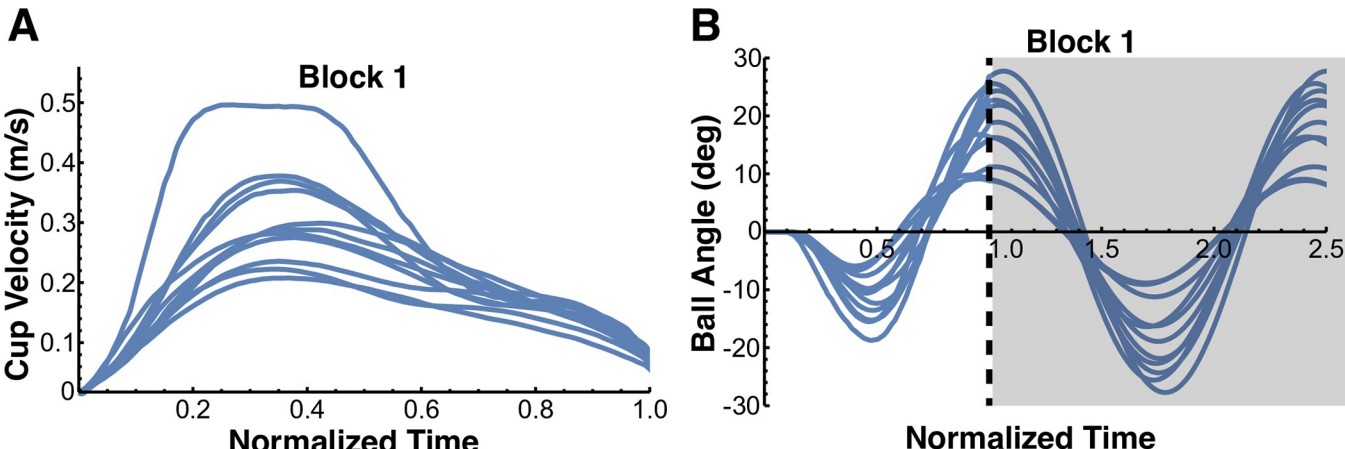

**Fig 4. A.** Mean velocity profiles across all trials for all 11 subjects in block 1. The velocity profiles of all subjects displayed an asymmetric profile with one peak with an additional 'shoulder' caused by the moving ball acting upon the cup. **B.** Mean ball angle profiles across all trials of block 1 for every subject. As expected, residual ball angles were large after completion of the cup movement, since the subjects did not seek to minimize ball oscillations. The gray shaded area indicates the time after completion of the cup movement.

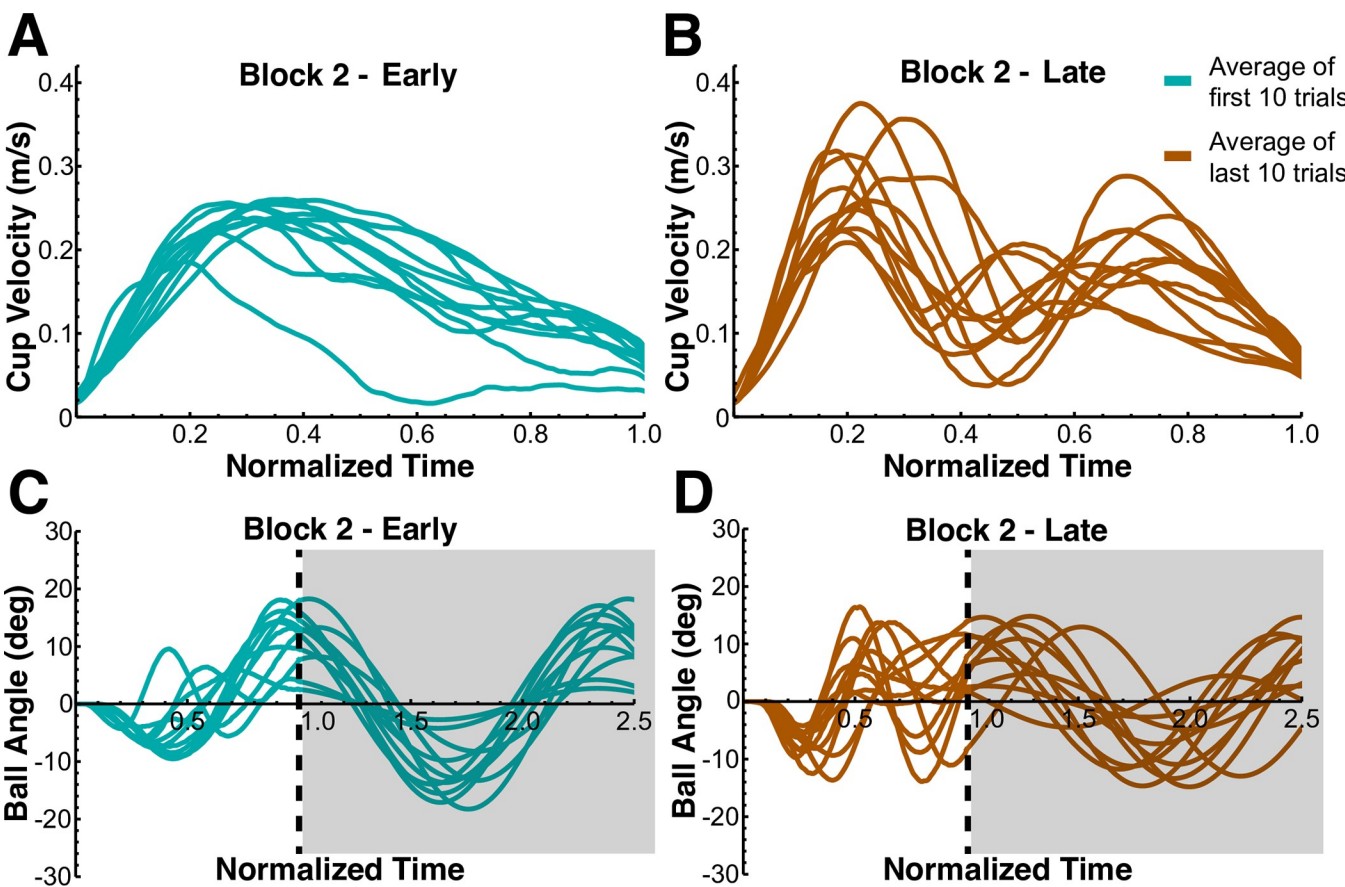

**Fig 5. Mean velocity and ball angle profiles across first and last 10 trials of block 2 for every subject. A.** At the beginning of the block, the velocity profiles of the subjects' movements exhibited a single peak with an extended decrease, similar to what was seen in block 1. **B.** By the end of the block all subjects had adopted a new pattern that displayed two peaks in the velocity profile. **C.** At the beginning of the block, subjects' movements were still exhibiting large residual oscillations, due to their single peak velocity profile. **D.** By the end of the block, all subjects were able to learn how to minimize residual oscillations to a satisfactory degree, due to the two-peak movement strategy. The gray shaded areas indicate the time after completion of cup movement.

confidence interval for the regression slope for all these 8 subjects. Only 3 subjects exhibited improvements with a statistically significant negative trend, indicating overall performance improvement with practice.

**Table 1. Regression statistics for residual ball angle vs trial number in block 2.** Asterisks highlight the subjects with a significantly negative regression slope.

| Subject | Regression Slope | 95% CI | *p*-value |
| --- | --- | --- | --- |
| S1 | -0.30 | [-0.45, -0.16]* | < 0.001 |
| S2 | 0.08 | [-0.13, 0.29] | 0.44 |
| S3 | -0.11 | [-0.23, 0.008] | 0.07 |
| S4 | 0.11 | [-0.06, 0.28] | 0.19 |
| S5 | 0.06 | [-0.07, 0.18] | 0.35 |
| S6 | -0.13 | [-0.26, -0.002]* | 0.046 |
| S7 | -0.01 | [-0.15, 0.12] | 0.87 |
| S8 | -0.09 | [-0.34, 0.16] | 0.49 |
| S9 | -0.08 | [-0.25, 0.08] | 0.30 |
| S10 | -0.15 | [-0.31, 0.002] | 0.053 |
| S11 | -0.25 | [-0.36, -0.14]* | < 0.001 |

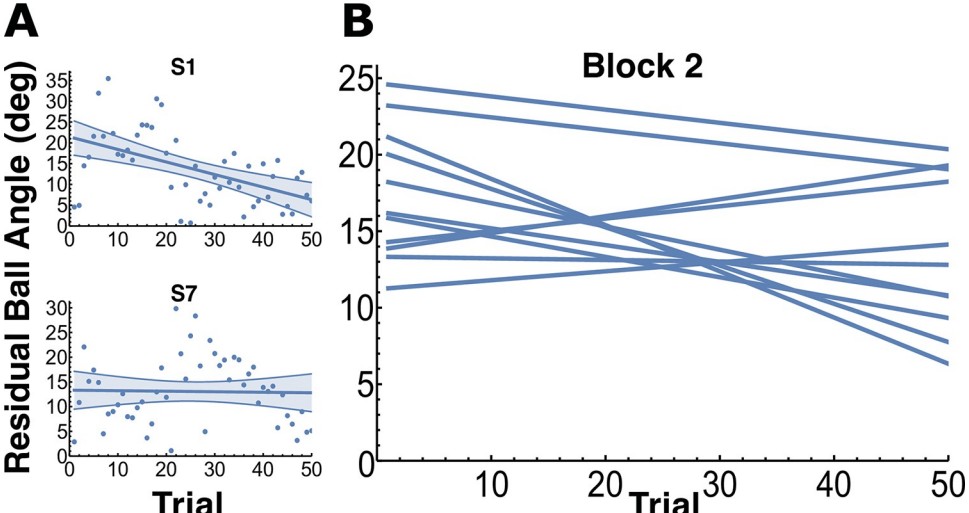

**Fig 6. Residual ball angles vs trial number for block 2 where subjects were instructed to focus on minimizing the residual ball oscillations. A.** The two panels on the left display two representative subjects' data (S1 with a negative trend and S7 with no trend) along with the regression line. The shaded bands represent the 95% confidence interval for the slope. **B.** The panel on the right displays the regression lines for all 11 subjects (statistics in Table 1). Eight subjects did not show any statistically significant trends with practice, which testifies that the task was difficult. The remaining 3 subjects had statistically significant negative trends, with the regression slope confidence intervals not containing zero.

## Testing blocks

In blocks 3 and 4, subjects were required to both minimize residual ball oscillations and complete the movement within a specified amount of time (1.40 s for block 3 and 1.20 s for block 4). The mean movement durations across all subjects was 1.50 ± 0.25 s in block 3 and 1.36 ± 0.20 s in block 4, which were longer than the desired movement durations in each block, reflecting the difficulty of performing the task quickly. However, the specified movement duration was not a hard constraint; it was only intended to deflect the subjects from the trivial solution of moving very slowly and to ensure sufficiently fast movements to elicit a dynamic ball behavior. With that in mind, subjects satisfied the movement duration constraint to an acceptable degree.

The mean velocity and ball angle profiles for every subject across all trials of blocks 3 and 4 are shown in Fig 7. The two-peaked velocity profiles that subjects adopted towards the end of the second familiarization block were maintained and became more regular in block 3 and block 4. The ball kinematics displayed small residual oscillations (compare to block 1, Fig 4B and block 2, Fig 5C and 5D), indicating that subjects performed well in the task. Fig 8 further displays the mean value of the residual ball angle, achieved by every subject in the testing blocks 3 and 4. The mean residual ball angle across all subjects was 10.39 ± 5.60 deg in block 3 and 11.38 ± 5.60 deg in block 4, with six subjects achieving a mean residual ball angle ≤ 10 deg in block 3 and four subjects achieving a mean residual ball angle ≤ 10 deg in block 4. As the instructed residual ball angle was ≤ 10 deg, these values indicated that subjects were successful in minimizing the residual ball angle. Altogether, these results show that the subjects were able to meet the task instructions of minimizing residual ball angles while moving at the required pace, hence performing the task with a reasonable level of skill.

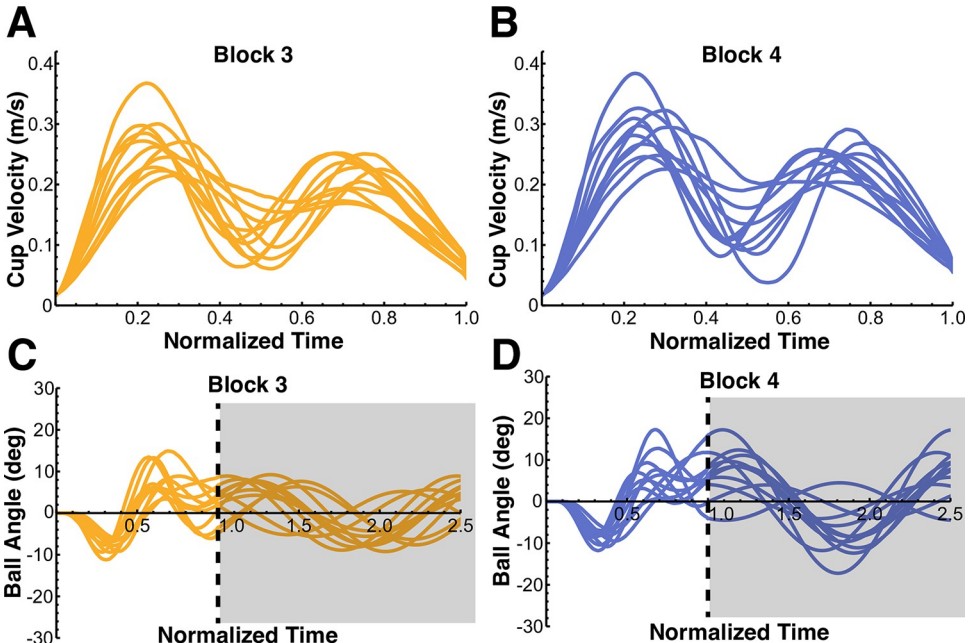

**Fig 7. Mean velocity and ball angle profiles across all trials of block 3 and block 4 for every subject. A,B.** In blocks 3 and 4, subjects maintained the two-peak velocity profiles that they learned in block 2. An important observation is the inequality in the peaks: the first peak was consistently higher than the second peak. **C,D.** Subjects were able to achieve small residual ball oscillations in blocks 3 and 4 (compare to Figs 4B, 5C and 5D). The gray shaded area indicates the time after completion of the cup trajectory.

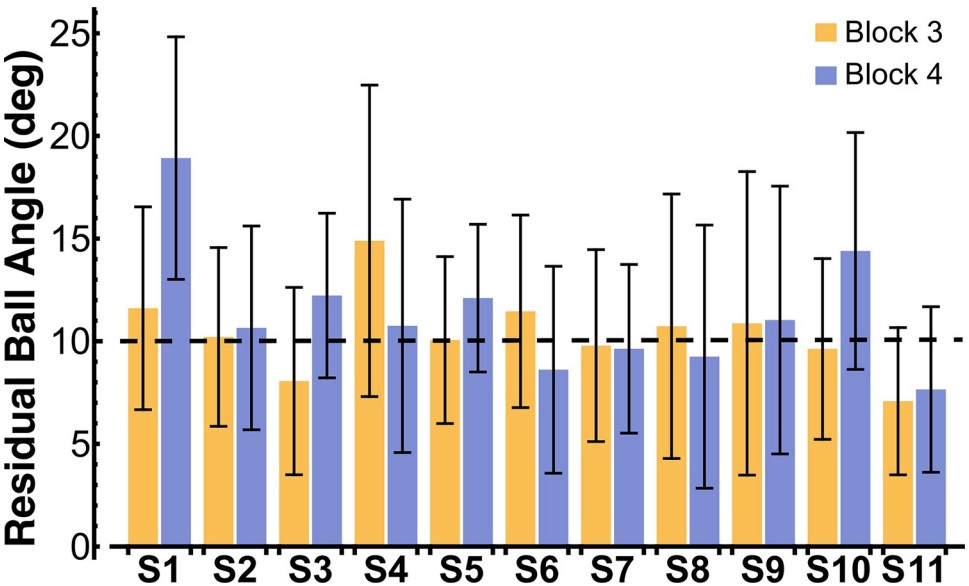

**Fig 8. Mean residual ball angles achieved by every subject in blocks 3 and 4.** The error bars denote one standard deviation. Subjects were instructed to aim for a residual ball angle ≤ 10 deg, which is marked by the dashed line in the figure.

### Hypothesis 1: Two peaks in the velocity profiles

As was pointed out earlier, the first robust observation in the velocity profiles in blocks 3 and 4 was the presence of two peaks in the velocity profile (Fig 7A and 7C). This matches the number of peaks predicted from the simulations presented in Fig 3.

### Hypothesis 2: Symmetric peaks in the velocity profiles

Contrary to the simulations however, the two peak amplitudes were not equal, with the tendency of the first peak to be higher than the second peak. Defining the peak ratio as $\frac{peak\ 1}{peak\ 2}$, 8 out of the 11 subjects had a mean peak ratio $> 1$ in block 3, while 9 subjects had a mean peak ratio $> 1$ in block 4 (Fig 9). The mean peak ratio across all subjects was $1.17 \pm 0.30$ in block 3 and $1.13 \pm 0.25$ in block 4. These means were significantly different from 1 in both blocks: block 3: $t(10) = 3.10$, $p = 0.01$; block 4: $t(10) = 3.11$, $p = 0.01$. This observation was not consistent with the simulation predictions based on a precise internal model of the linearized dynamics (Fig 3). While the imprecision and inequality of peak values was not surprising, it was not of a random nature, but rather exhibited systematic features. There was robust signature of "imprecision" whereby the first peak was larger than the second peak. This salient behavioral feature served as a guide in further investigations below.

### Hypothesis 3: Positive correlation between inter-peak minimum velocity and movement duration

Counter to expectations, the correlations between the inter-peak minimum velocity and the movement duration were negative. Out of the 11 subjects, 4 subjects in block 3 and 7 subjects in block 4 displayed a behavior that resulted in a statistically significant negative regression line between the inter-peak minimum velocity and movement duration, i.e., smaller inter-

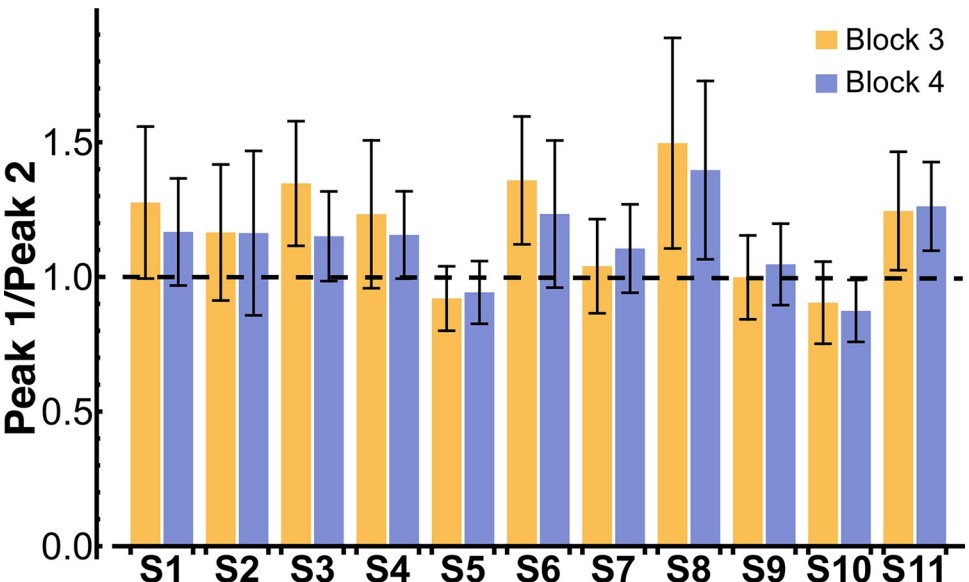

**Fig 9. Mean peak ratio across all trials of block 3 and block 4 for every subject.** The error bars denote one standard deviation. In block 3, eight subjects had a mean peak ratio $> 1$, while in block 4 nine subjects had a mean peak ratio $> 1$.

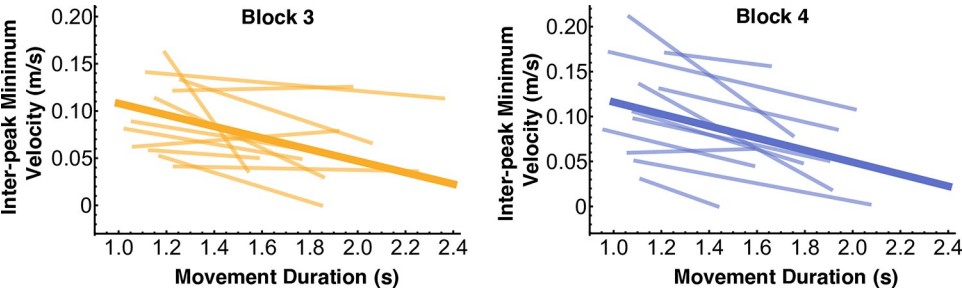

**Fig 10. The lighter-colored lines show the linear regressions between the inter-peak minimum velocity and the movement duration for every subject in blocks 3 and 4 (for statistics, refer to Table 2).** The thick lines are the fitted linear mixed-effects models, which show a significant negative main effect of movement duration on inter-peak velocity.

peak minimum velocities for longer movement durations (Fig 10, light-colored lines). The zero slope was in the 95% confidence interval for all the remaining subjects, indicating no statistically significant relation, i.e., no subject displayed a positive relation (Table 2). A linear mixed-effects model with subject as a random factor was fitted in block 3 and block 4 (Fig 10, thick lines). In both blocks, movement duration had a significant negative main effect on inter-peak velocity (block 3: $\beta = -0.061 \pm 0.026, p < 0.001$; block 4: $\beta = -0.067 \pm 0.029, p < 0.001$). This negative relation between inter-peak velocity and movement duration also contradicted the simulation predictions portrayed in Fig 3. For comparison with Fig 3, the cup velocity profiles of two representative subjects are displayed in Fig 11, color-coded by movement duration.

## Simulations: Testing different internal models

The experimental results showed consistent patterns with two peaks in the velocity profile as predicted by an input shaping strategy (Hypothesis 1, Fig 3). However, the two peaks had different amplitudes, counter to the predictions for input shaping when applied to the system dynamics described in Eqs (4)–(5) (Hypothesis 2). Also counter to Hypothesis 3, at the minimum between the two peaks the velocity decreased with movement duration, not increased as predicted. We therefore concluded that subjects implemented an input shaping control strategy, but that the assumed internal model required modification.

**Table 2. Regression statistics for inter-peak minimum velocity vs movement duration in blocks 3 and 4.** Asterisks highlight significantly negative regression slopes.

| Subject | Block 3 | | | Block 4 | | |
|---|---|---|---|---|---|---|
| | Regression Slope | 95% CI | p | Regression Slope | 95% CI | p |
| S1 | -0.36* | [-0.48, -0.24] | < 0.001 | -0.06 | [-0.13, 0.002] | 0.06 |
| S2 | -0.04 | [-0.10, 0.02] | 0.18 | -0.05* | [-0.09, -0.008] | 0.02 |
| S3 | -0.02 | [-0.07, 0.03] | 0.40 | -0.03 | [-0.11, 0.04] | 0.36 |
| S4 | -0.08* | [-0.13, -0.03] | 0.003 | -0.10* | [-0.15, -0.04] | < 0.001 |
| S5 | -0.12* | [-0.17, -0.07] | < 0.001 | -0.15* | [-0.20, -0.10] | < 0.001 |
| S6 | -0.005 | [-0.03, 0.02] | 0.69 | -0.06* | [-0.10, -0.01] | 0.01 |
| S7 | -0.04 | [-0.10, 0.01] | 0.12 | -0.08* | [-0.13, -0.03] | 0.002 |
| S8 | -0.02 | [-0.15, 0.12] | 0.78 | -0.06 | [-0.20, 0.07] | 0.33 |
| S9 | 0.02 | [-0.05, 0.09] | 0.55 | 0.008 | [-0.08, 0.09] | 0.85 |
| S10 | 0.006 | [-0.05, 0.06] | 0.85 | -0.19* | [-0.25, -0.13] | < 0.001 |
| S11 | -0.08* | [-0.12, -0.05] | < 0.001 | -0.06* | [-0.12, -0.005] | 0.03 |

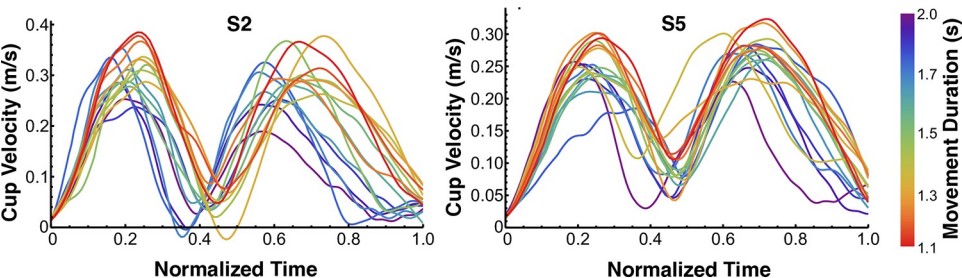

**Fig 11. Velocity profiles for two representative subjects S2 and S5 in block 4.** As opposed to the simulation predictions in Fig 3, the inter-peak minimum velocities increased with shorter movement durations.

## Hand-object coupled dynamics model

Given that subjects were instructed to reduce residual ball oscillations in the experiment, input shaping was used as a template for possible control strategies they employed. For the cup-and-ball system in Eqs (4)–(5) the damping ratio was zero and, hence, the damped natural period of oscillation $T_d$ was equal to the natural period of the assumed pendulum $T_n = 2\pi\sqrt{\frac{l}{g}}$.

Assuming a perfect internal model, from Eq (3), the required amplitudes and timings of the impulses were computed to be $A_1 = 0.5$, $A_2 = 0.5$, $t_1 = 0$, $t_2 = 0.71$. Simulations using this shaped input showed that this model predicted cup velocity profiles with equal peaks (Fig 3). Further, this model predicted that for faster movements, the inter-peak minimum velocity should decrease. Both predictions were in stark contrast to the experimental observations, indicating a fundamental component was missing.

The decrease in amplitude in the second peak in the data suggested that the biological system was dissipating energy, in contrast to the system modeled in Eqs (4)–(5). As the virtual system did not have any damping, the hand-object coupling must have played a significant role in the subjects' performance. This meant that the system's equations of motion needed to include hand interactive dynamics coupled to the cart-and-pendulum system. This was achieved by adding an impedance element, modeled as an ideal force generator ($F_{ff}$) in parallel with a spring (stiffness $K$) and a damper (damping coefficient $B$) to the cart-and-pendulum system (Fig 12). $F_{ff}$ denotes the feedforward force required to follow a desired trajectory $[x_{des}(t), \dot{x}_{des}(t)]$. If the full dynamics of the cart-and-pendulum system were perfectly anticipated, including the ball's perturbing forces, subjects would be able to generate an $F_{ff}$ that would allow the cart to follow the desired trajectory exactly, i.e., $x = x_{des}$ and $\dot{x} = \dot{x}_{des}$. Due to the system's complex dynamics, however, it is unlikely that subjects learned the precise model. If, instead, the internal model differed from the system's dynamics, the spring and damper would provide restoring forces back toward the desired trajectory. Adding the feedforward force and impedance element to Eqs (4)–(5) results in these equations of motion:

$$(m_c + m_p)\ddot{x} = -m_p l\ddot{\phi} + F \tag{6}$$

$$\ddot{\phi} = -\frac{g}{l}\phi - \frac{\ddot{x}}{l} \tag{7}$$

$$F = F_{ff} + B(\dot{x}_0 - \dot{x}) + K(x_0 - x) \tag{8}$$

The impedance parameters $K$ and $B$ were considered constant during a single trial. The

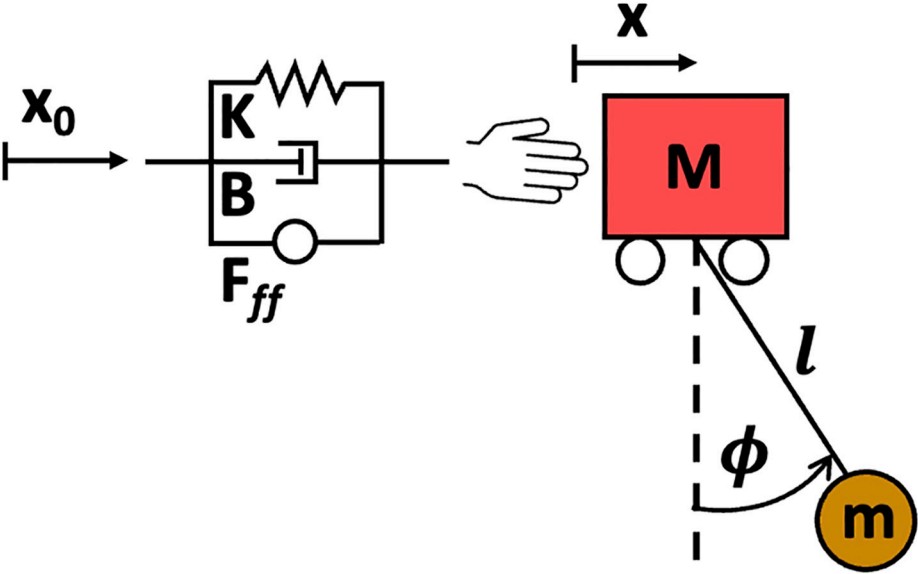

**Fig 12. Coupled hand-object system used to model the task in simulations.** The dynamics of the cart-pendulum system were coupled to a model of hand impedance. $x_0$, shaped cart position; $K$, hand stiffness; $B$, hand damping coefficient; $F_{ff}$, feedforward force; $x$, cart position; $M$, mass of the cart; $l$, pendulum length; $\phi$, pendulum angle; $m$, mass of the pendulum.

coupled system (6)–(8) was a 4<sup>th</sup>-order dynamical system, which meant that it exhibited two modes of oscillation with different decay rates [23]. How would subjects plan and shape the inputs $[x_0, \dot{x}_0]$? Were subjects accounting for the hand impedance and, hence, planning their inputs based on an accurate model of the full 4<sup>th</sup>-order cart-and-pendulum system coupled to the hand? Or were their inputs planned and generated based on a simpler, lower-order internal model? To gain insight into the internal representation that subjects may have developed of the coupled hand plus cart-and-pendulum system, input shaping controllers with varying internal models were developed and compared with the data.

Five different internal models were considered to generate shaped inputs as illustrated in Fig 13. In each case, the impulse amplitudes and timings were computed based on the selected internal model and convolved with a minimum-jerk profile as before. The resulting shaped input $[x_0, \dot{x}_0, \ddot{x}_0]$ was delivered to the corresponding internal model and forward simulated to generate the desired kinematic profile $[x_{des}, \dot{x}_{des}, \ddot{x}_{des}]$ and the feedforward force $F_{ff}$. Finally, $F_{ff}$ and the shaped input were applied to a simulation of the nonlinear equations of motion including hand impedance:

$$(m_c + m_p)\ddot{x} = m_p l(\dot{\phi}^2 \sin\phi - \ddot{\phi} \cos\phi) + F \tag{9}$$

$$\ddot{\phi} = -\frac{\ddot{x}}{l} \cos\phi - \frac{g}{l} \sin\phi \tag{10}$$

$$F = F_{ff} + B(\dot{x}_0 - \dot{x}) + K(x_0 - x) \tag{11}$$

For all simulations, the shaped input was computed based on a different hypothesized linear internal model of (6)-(8). The shaped input was always entered into that internal model to generate a desired trajectory as an output. Finally, that desired trajectory was used to calculate a

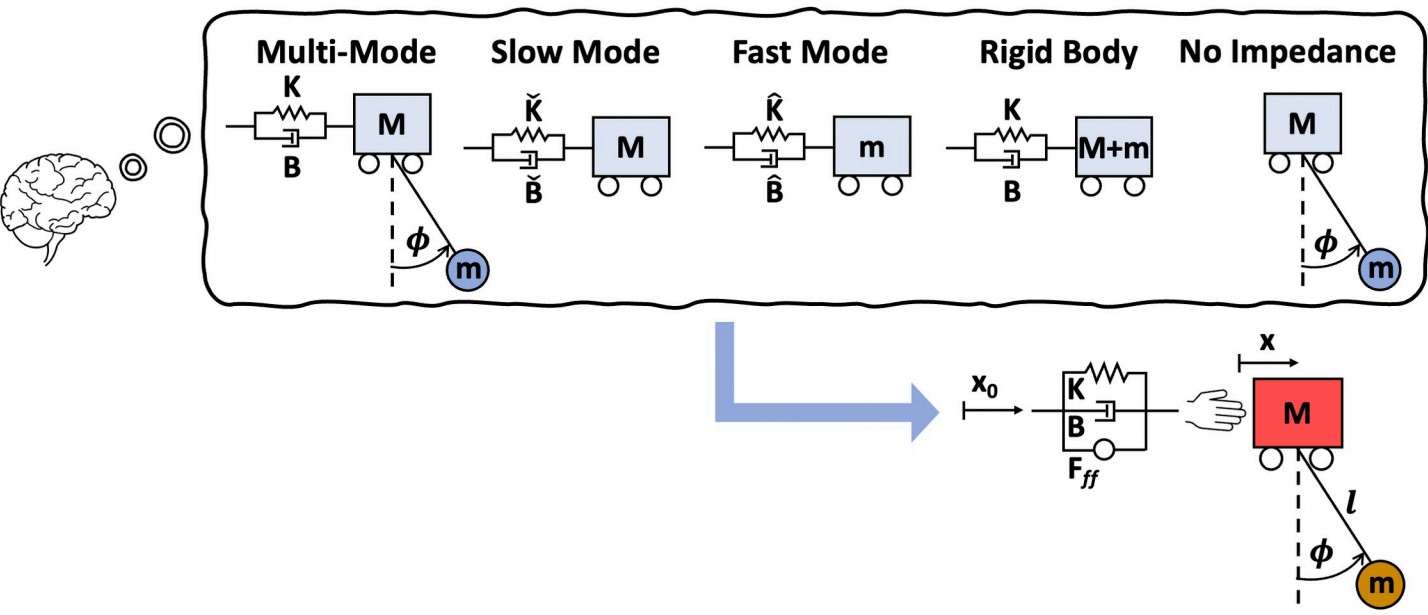

**Fig 13. The five different internal models used to generate control inputs to the system via input shaping.** Once $x_0$ and $F_{ff}$ were generated depending on the internal model, they were fed into the coupled nonlinear system (Eqs 9–11) and simulated.

feedforward force $F_{ff}$ which was entered into the fully fledged coupled nonlinear system (9)-(11) along with the shaped input $[x_0, \dot{x}_0]$. This simulation process is visualized in Fig 13 and summarized in a flow chart in the Supplementary Material S1 Fig.

After careful study of the object's dynamics, and through systematic reduction, five candidate internal models were identified. These models were a full set, insofar as they included all structural simplifications of this system, which has only two dynamic modes of behavior. Full derivations of the feedforward force generated based on each internal model are provided in the Supplementary Material S1 Text. The five different internal models were:

### I. Internal model of the coupled system: Multi-Mode (MM)

The first model assumed that subjects developed a precise internal model of the 4th-order coupled system and sought to cancel both modes of oscillation. For systems with multiple modes that each contribute significantly to a system's behavior, a multi-mode input shaper was needed to successfully move the system from point to point with zero residual oscillations. A simple implementation of a multi-mode input shaper was the convolved shaper, achieved by convolving together input shapers for each individual mode [24]. The resultant multi-mode shaper maintained the oscillation-cancelling properties of each individual shaper. In this case of a two-mode shaper, a total of four impulses were required to successfully achieve the task.

To generate the feedforward force input for this model, Eqs (6)–(8) were integrated using the shaped input $x_0$ and $\dot{x}_0$, producing desired cart-and-pendulum kinematics, $[x_{des}, \dot{x}_{des}, \ddot{x}_{des}]$ and $[\phi_{des}, \dot{\phi}_{des}, \ddot{\phi}_{des}]$. Combining and rearranging Eqs (6)–(8), the feedforward force $F_{ff}$ was hence calculated as

$$F_{ff} = m_c \ddot{x}_{des} - m_p g \phi_{des} \tag{12}$$

## II. Internal model of the coupled system: Slow Mode (SM)

Similar to the Multi-Mode model, the Slow Mode model assumed that subjects had a precise internal model of the $4^{\text{th}}$-order coupled system. However, this model assumed that subjects sought to cancel only one of the oscillation modes, specifically the lower-frequency mode, requiring only two input shaping impulses. This model could have represented a control strategy in which subjects, faced with a difficult task, attempted to cancel only the slower oscillation in the system, which may have been easier to do.

A single vibratory mode of a system with multiple degrees of freedom could be represented by an equivalent mass-spring-damper system. The natural frequency and damping ratio of the slower mode were first calculated. The effective modal mass was set equal to that of the larger of the two masses in the system, the cup. The effective modal stiffness and damping of this equivalent system were then calculated as

$$\tilde{K}_{slow} = m_c \omega_{slow}^2 \tag{13}$$

$$\tilde{B}_{slow} = 2m_c \zeta_{slow} \omega_{slow} \tag{14}$$

where $\omega_{slow}$ and $\zeta_{slow}$ are the natural frequency and damping ratio of the slower mode, respectively. A shaped input for this internal system was generated by inserting the damping ratio and damped natural period into Eq (3) and convolving the resulting impulses with the nominal minimum jerk profile. The shaped input $[x_0, \dot{x}_0]$ was then applied to the equation of motion of the equivalent mass-spring-damper system:

$$m_c \ddot{x} + \tilde{B}_{slow}(\dot{x} - \dot{x}_0) + \tilde{K}_{slow}(x - x_0) = F \tag{15}$$

Integrating Eq (15) produced the desired cup kinematics. The feedforward force input $F_{ff}$ was hence calculated as:

$$F_{ff} = m_c \ddot{x}_{des} \tag{16}$$

## III. Internal model of the coupled system: Fast Mode (FM)

The Fast Mode model was conceived in the same spirit as the slow mode model, with the difference that input shaping impulses were generated to cancel only the faster mode. This model could have represented a control strategy in which subjects focused only on cancelling the higher-frequency oscillation in the system while ignoring the lower-frequency oscillation. Such a strategy may have been encouraged by the experiment's emphasis on performing faster movements and minimizing pendulum motion.

An effective mass-spring-damper system was formulated in the same way as Eqs (13)–(15) but by using the natural frequency and damping ratio of the higher-frequency mode, as well as the smaller of the two masses in the system, i.e., that of the pendulum, $m_p$. Feedforward force input $F_{ff}$ was calculated similarly to Eqs (15)–(16):

$$F_{ff} = m_p \ddot{x}_{des} \tag{17}$$

## IV. Simplified internal model: Rigid Body (RB)

This model considered a simplified internal model that disregarded the pendulum's oscillatory dynamics, grouping the cart-and-pendulum to a single mass, resulting in a mass-spring-

damper system. As humans are adept at manipulating rigid objects, the motivation for this rigid body model was that subjects internally represented the total mass of the object and their own hand impedance. Such dynamics would be more familiar than the pendulum's internal degree of freedom. This internal model had the following characteristics, which were used to calculate the required impulses:

$$\zeta = \frac{B}{2\sqrt{K(m_c + m_p)}}, \omega = \sqrt{\frac{K}{(m_c + m_p)}}, T_d = \frac{2\pi}{\omega\sqrt{1-\zeta^2}} \tag{18}$$

Feedforward force was calculated as

$$F_{ff} = (m_c + m_p)\ddot{x}_{des} \tag{19}$$

## V. Simplified internal model: No Impedance (NI)

This model considered that subjects planned the shaped inputs based on the cart-and-pendulum dynamics expressed in Eqs (4)–(5), without regard to hand impedance. The impulse amplitudes and timings were therefore identical to those computed for the nominal input shaping model presented in Fig 3. Note however that this simulation differed from the input shaping simulation of Fig 3. The internal model without impedance was used to generate a feedforward force $F_{ff}$ that was fed as an input to the coupled system with impedance (9)-(11) rather than the system without impedance (4)-(5), producing different outputs. This No Impedance model assumed that subjects were mainly focused on the dynamics of the cart-and-pendulum system and did not consider the effect of their own hand impedance.

To generate the desired cart-and-pendulum trajectories, Eqs (4)–(5) were forward-simulated using the shaped input to the cup, $\ddot{x}_0$. Feedforward force $F_{ff}$ was then calculated as

$$F_{ff} = m_c\ddot{x}_{des} - m_p g\phi_{des} \tag{20}$$

## Model fitting using optimization

S1 and S2 Figs illustrate the process used to find optimal impedance values for each experimental trial using different internal models. For each of the five models, the desired cart kinematics $[x_{des}, \dot{x}_{des}, \ddot{x}_{des}]$ were computed based on the respective internal model, leaving three free parameters to fit a simulated trial to one experimental trial: stiffness $K$, damping $B$, and time added to the experimental trial duration $T$. To find the parameter values that resulted in the best fit to the experimental observations, a nonlinear optimization was conducted utilizing the open-source NLopt package [25]. The controlled random search algorithm (CRS2) with the "local mutation" modification was used for all optimizations [26].

Hand impedance values were allowed to vary over a large range: damping values were bounded from 0 to 100 Ns/m and stiffness values were bounded from 0 to 1000 N/m. Initial values were chosen as $B = 10$ Ns/m and $K = 100$ N/m, reasonable order-of-magnitude estimates for upper arm impedance [14]. The optimization also included additional trial duration $T$ as a parameter, which was allowed to vary from 0 to 1000 ms. This was a necessary step to simulate longer trials starting from rest before they were trimmed at non-zero velocity start and end thresholds, enabling comparison to the experimental trials.

The objective function for the nonlinear optimization was defined as the root mean square error (RMSE) between simulated and observed kinematic profiles at all time points. Both

profiles were defined using weighted kinematic variables normalized to that variable's range. RMSE for one kinematic variable was defined as:

$$RMSE_x = \frac{\sqrt{w_x \sum_{i=1}^{N} \left( x(t_i) - \hat{x}(t_i) \right)^2}}{\max(x) - \min(x)} \tag{21}$$

where $w_x$ is the weight assigned to variable $x$. The total RMSE was then calculated by summing the RMSE for each variable and dividing by the sum of the square roots of the weights. Further details on the optimization methods are included in S1 Text. The dynamic simulation and optimization code used in this study may be found at https://github.com/stephan-stansfield/cup-task.

## Simulation results

The simulated data were analyzed in the same way as the experimental data. The criteria for evaluating the adequacy of the model were the two robust features in the behavioral data: the unequal peaks in the velocity profiles and the negative relation between inter-peak minimum velocity and movement duration. In addition, the variance of the cup velocity accounted for by the simulation along with the best-fit impedance values (selected by the optimization algorithm) were used to assess model competence.

### Unequal peaks in the simulated velocity profiles

All five candidate models had a mean velocity peak ratio that was greater than 1 (Fig 14). The interaction between Block and Model was significant ($\chi^2(5) = 34.57$, p<0.0001). Planned pairwise comparisons with Bonferroni corrections between the experimental mean peak ratio and the mean peak ratios for the different models revealed that block 3 of the Fast Mode (FM) model, and block 4 of the Multi-Mode (MM) model were significantly different from the experimental (Exp) data (block 3: Exp vs FM $t(5766) = -2.583, p = 0.05$; block 4: Exp vs MM $t(5766) = -2.609, p = 0.05$) (Fig 14 and Table 3). Hence, this first test gave an indication that the Multi-Mode and Fast Mode models did not replicate the subjects' behavior.

### Inter-peak minimum velocity vs movement duration correlation

To investigate which simulated models reproduced the negative relation between movement duration and inter-peak minimum velocity, we first fitted a linear mixed-effects model with Model, Block, Movement Duration and their interactions as the main predictors. The interaction between Block, Model and Movement Duration was significant ($\chi^2(5) = 22.56$, p = 0.0004) (Fig 15). Planned pairwise comparisons with Bonferroni corrections between the experimental data and the simulated models revealed that only the Fast Mode and Rigid Body models were not significantly different from the experiment in both blocks, while the Multi-Mode model was not significantly different only in block 3 (Table 4). Taken together with the peak ratio results, the Rigid Body model was the only model that was capable of reproducing both experimental observations in both blocks.

### Variance Accounted For (VAF)

As shown above, the simulated models produced movement kinematics which fit the subjects' movements to varying degrees. To further evaluate the model fits, the cup velocity variance

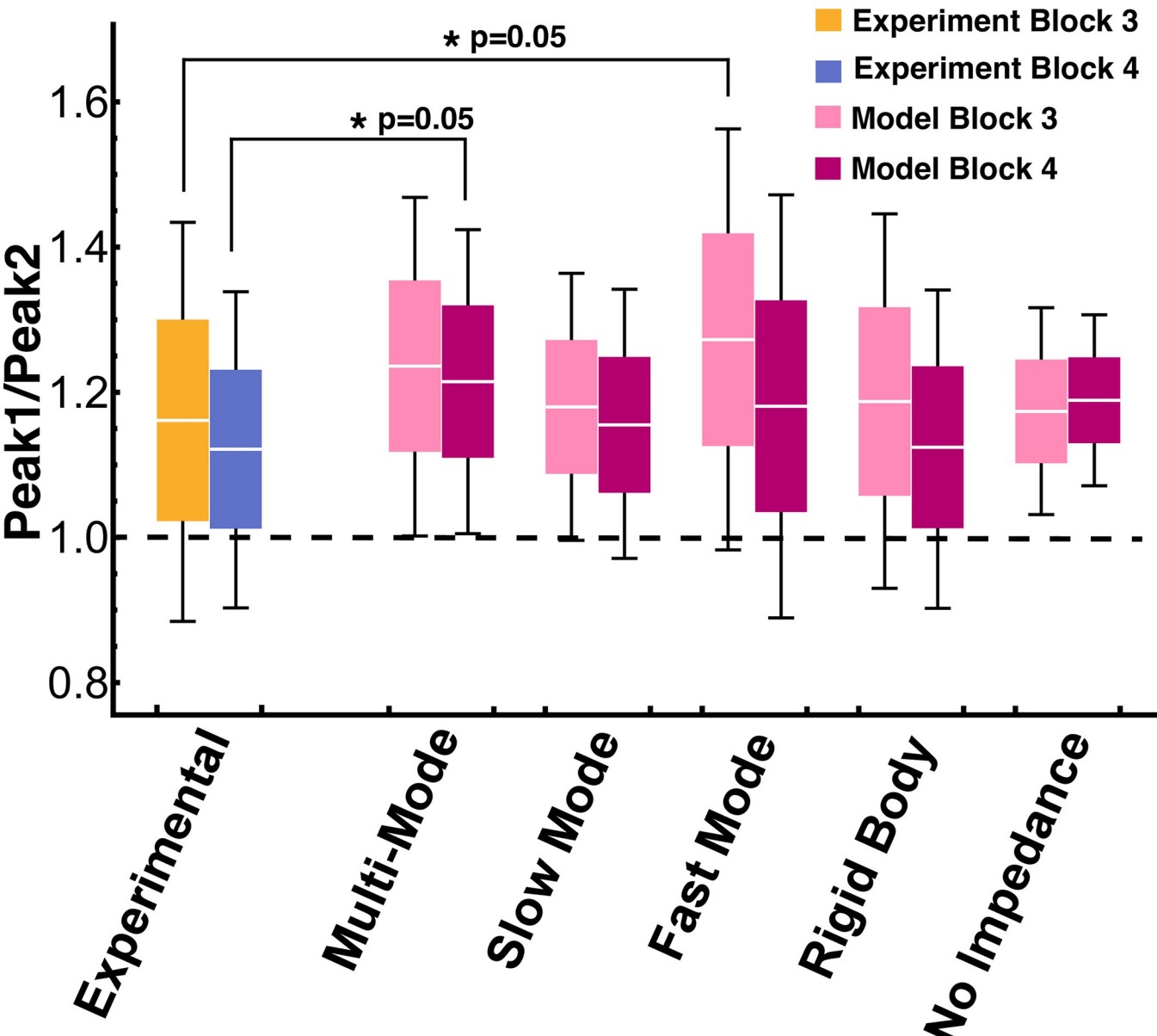

**Fig 14. Velocity peak ratios from experimental and simulated trials of block 3 and block 4 for the five different models.** The white line in every box denotes the mean, the box height is half a standard deviation, and the error bars denote one standard deviation. Planned pairwise comparisons between the experimental mean peak ratio and the mean peak ratios of the models revealed that block 3 of the Fast Mode model and block 4 of the Multi-Mode model were statistically significantly different from the experimental means (Table 3). This was the first indication that the Multi-Mode and Fast Mode models did not match the data.

accounted for (VAF) for each trial was calculated as

$$VAF = 100\% * max\left(1 - \frac{\sum_{i=1}^{N}\left[v(t_i) - \hat{v}(t_i) - \bar{v} + \bar{\hat{v}}\right]^2}{\sum_{i=1}^{N}\left[v(t_i) - \bar{v}\right]^2}, 0\right) \tag{22}$$

where N was the number of data points analyzed in a trial, $v(t_i)$ was the experimentally observed cup velocity at one time point, $\hat{v}(t_i)$ was the simulated cart velocity at the same time

**Table 3. Planned pairwise comparisons between the mean velocity peak ratio of the experimental data and simulated models.** Asterisks highlight statistically significant differences.

| Comparison | Block 3 | | | | Block 4 | | | |
|---|---|---|---|---|---|---|---|---|
| | Mean Difference | SE | *t* | *p* | Mean Difference | SE | *t* | *p* |
| Exp vs MM | -0.06 | 0.03 | -1.8 | 0.33 | -0.08* | 0.03 | -2.6 | 0.05 |
| Exp vs SM | -0.02 | 0.03 | -0.67 | 1 | -0.03 | 0.03 | -0.98 | 1 |
| Exp vs FM | -0.08* | 0.03 | -2.6 | 0.05 | -0.04 | 0.03 | -1.2 | 1 |
| Exp vs RB | -0.02 | 0.03 | -0.57 | 1 | -0.01 | 0.03 | 0.44 | 1 |
| Exp vs NI | -0.01 | 0.03 | -0.38 | 1 | -0.06 | 0.03 | -2.1 | 0.18 |

point, and $\bar{v}$ and $\bar{\hat{v}}$ were the mean values of $v(t)$ and $\hat{v}(t)$ across the movement interval, respectively [27].

The mean and standard deviation of subjects' cart velocity VAF for each model in block 3 and block 4 are shown in Fig 16. The Rigid Body model produced the highest VAF (84 ± 14% in block 3 and 87 ± 10% in block 4), providing further evidence that it was the best-fit model. The interaction between Model and Block was significant ($\chi^2(4) = 100.95$, p < 0.0001). Planned pairwise comparisons with Bonferroni corrections between the Rigid Body model and the rest of the simulated models revealed that the mean VAF in the Rigid Body model was significantly different from all other models in both blocks, except for the Multi-Mode model in block 3 (Table 5).

## Best-Fit impedance values

To further evaluate the model fits, the best-fit impedance values were inspected. There was considerable variability in the best-fit impedance values between the different models (Fig 17). The best-fit stiffness and damping values for the Multi-Mode, Slow Mode, and No Impedance models varied widely across subjects. The ranges of subject mean best-fit stiffness values for

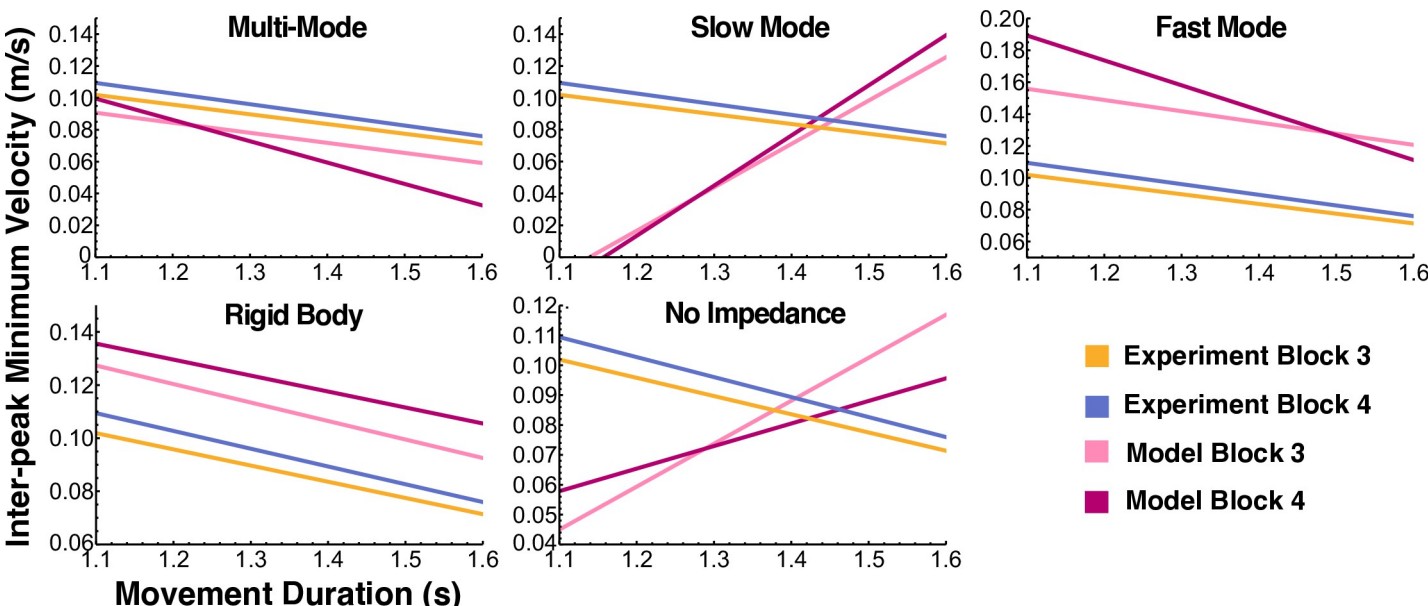

**Fig 15. Regression lines for inter-peak minimum velocity versus movement duration.** Planned pairwise comparisons between the experimental data and the simulated models revealed that only the Fast Mode and Rigid Body models replicated, in both blocks, the negative relation observed in the data (Table 4).

**Table 4. Planned pairwise comparisons between the regression lines of inter-peak velocity vs movement duration of the experimental data and simulated models.** Asterisks highlight statistically significant differences.

| Comparison | Block 3 | | | | Block 4 | | | |
|---|---|---|---|---|---|---|---|---|
| | Mean Difference | SE | t | p | Mean Difference | SE | t | p |
| Exp vs MM | -0.001 | 0.02 | -0.05 | 1 | 0.06* | 0.02 | 3.1 | 0.01 |
| Exp vs SM | -0.34* | 0.03 | -13.3 | <0.001 | -0.39* | 0.03 | -14.0 | <0.001 |
| Exp vs FM | 0.0008 | 0.03 | 0.03 | 1 | 0.05 | 0.02 | 2.0 | 0.2 |
| Exp vs RB | 0.007 | 0.02 | 0.35 | 1 | -0.005 | 0.02 | -0.25 | 1 |
| Exp vs NI | -0.21* | 0.02 | -9.2 | <0.001 | -0.14* | 0.02 | -6.1 | <0.001 |

these three models were greater than 330 N/m, while the ranges for the Fast Mode and Rigid Body models were less than 100 N/m. Likewise, the ranges of subject mean best-fit damping coefficients were greater than 30 N/m for Multi-Mode, Slow Mode, and No Impedance, and

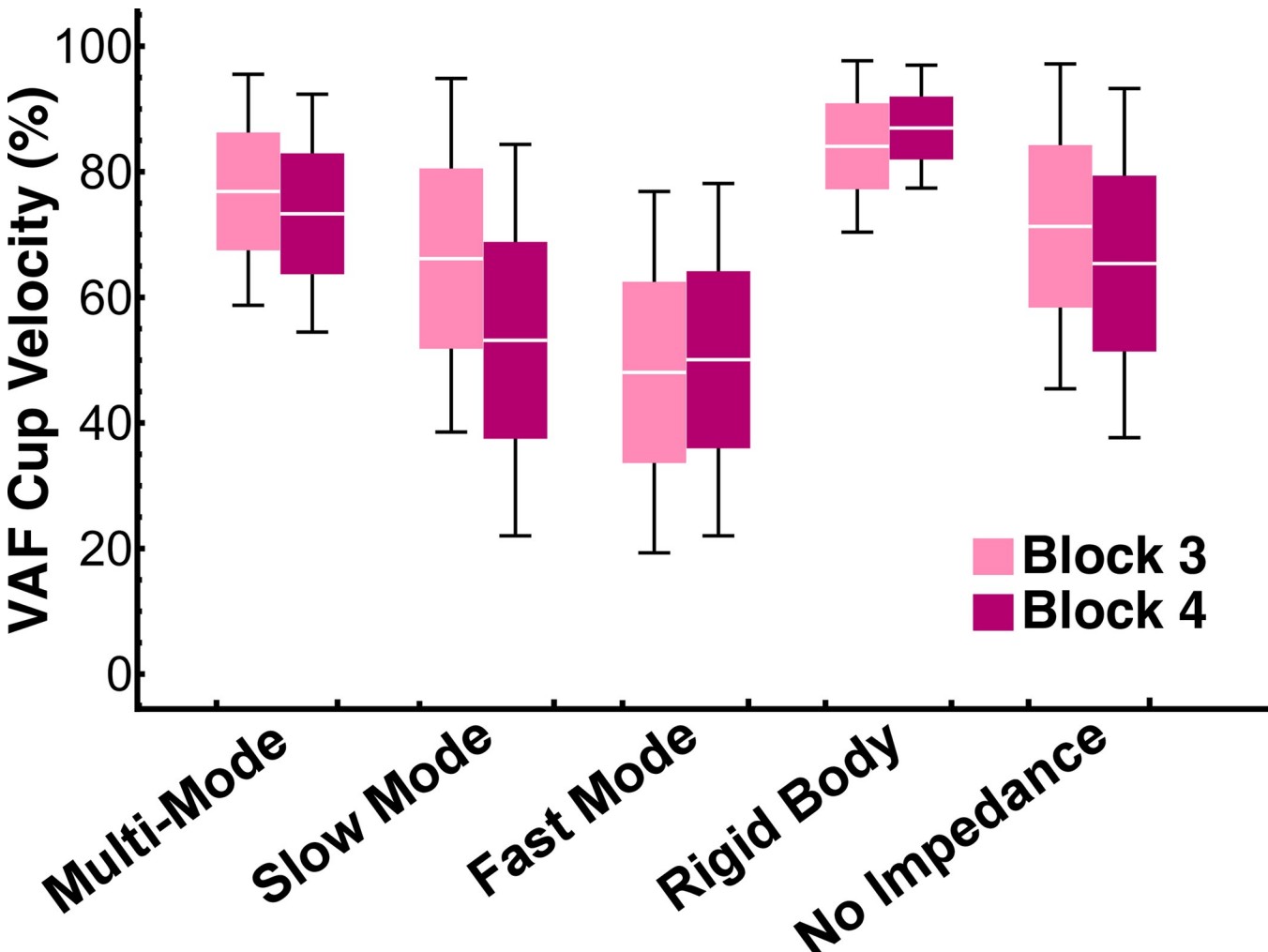

**Fig 16. Cup velocity variance accounted for (VAF) in block 3 and block 4 of every model.** The white line in every box denotes the mean, the box height is half a standard deviation, and the error bars denote one standard deviation. The Rigid Body model had the highest variance accounted for (84±14% in block 3 and 87±10% in block 4). Planned pairwise comparisons revealed that the mean VAF in the Rigid Body model was significantly different from all other models in both blocks, except for the Multi-Mode model in block 3 (Table 5).

**Table 5. Planned pairwise comparisons between the variance accounted for (VAF) of the Rigid body model (highest VAF) and the rest of the simulated models.** Asterisks highlight statistically significant differences.

| Comparison | Block 3 | | | | Block 4 | | | |
|---|---|---|---|---|---|---|---|---|
| | Mean Difference | SE | t | p | Mean Difference | SE | t | p |
| RB vs MM | -0.15 | 0.07 | -2.0 | 0.41 | -0.28* | 0.07 | -4.0 | <0.001 |
| RB vs SM | -0.33* | 0.07 | -4.6 | <0.001 | -0.61* | 0.07 | -8.6 | <0.001 |
| RB vs FM | -0.66* | 0.07 | -9.2 | <0.001 | -0.68* | 0.07 | -9.6 | <0.001 |
| RB vs NI | 0.24* | 0.07 | 3.3 | 0.01 | 0.40* | 0.07 | 5.7 | <0.001 |

were less than 11 Ns/m for Fast Mode and Rigid Body. For the Fast Mode model, the overall mean best-fit stiffness was 85 N/m in block 3 and 97 N/m in block 4, and the overall mean best-fit damping was 6 Ns/m in block 3 and 4.3 Ns/m in block 4. For the Rigid Body model, the overall mean best-fit stiffness was 125 N/m in block 3 and 154 N/m in block 4, and the overall mean best-fit damping was 3.3 Ns/m in block 3 and 3.7 Ns/m in block 4. The Fast Mode and Rigid Body best-fit impedance values were within the range of previously reported values of hand impedance for tasks involving interaction with the cup-and-ball system [8,14,16]. Therefore, the Fast Mode and Rigid Body models had the most physiologically plausible impedance values. Another interesting observation in the Rigid Body model was evidence of a negative exponential relation between trial duration and best-fit hand stiffness (Fig 18).

Table 6 summarizes the performance of the different simulation models with regard to the evaluation criteria. Altogether, the simulation results showed that the control strategy that used the simplified Rigid Body internal model coupled with hand impedance dynamics provided the best match to the experimental data: it replicated both experimental observations, had the highest VAF, and used physiologically plausible impedance values to fit the data. This provided evidence that when physically interacting with complex objects, humans used a control strategy that relied on a simple internal model of the object dynamics along with impedance to manage the differences between model predictions and reality.

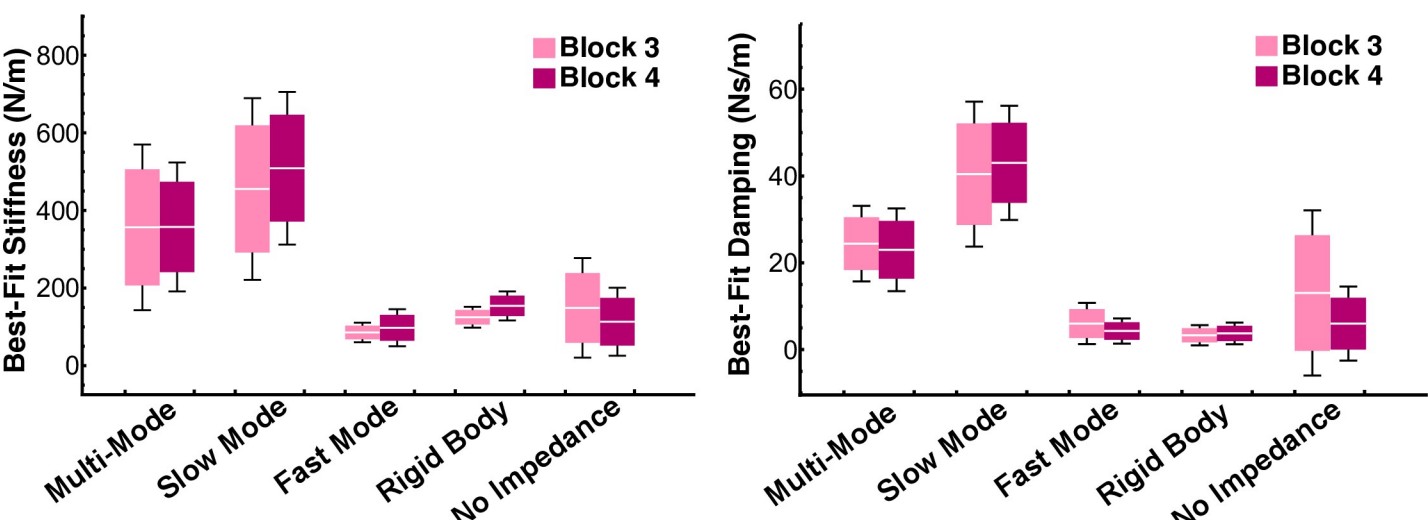

**Fig 17. Best-fit stiffness and damping values for each model.** The white line in every box denotes the mean, the box height is half a standard deviation, and the error bars denote one standard deviation. The Fast Mode and Rigid Body models had the most physiologically plausible impedance values, and the smallest inter-subject variability.

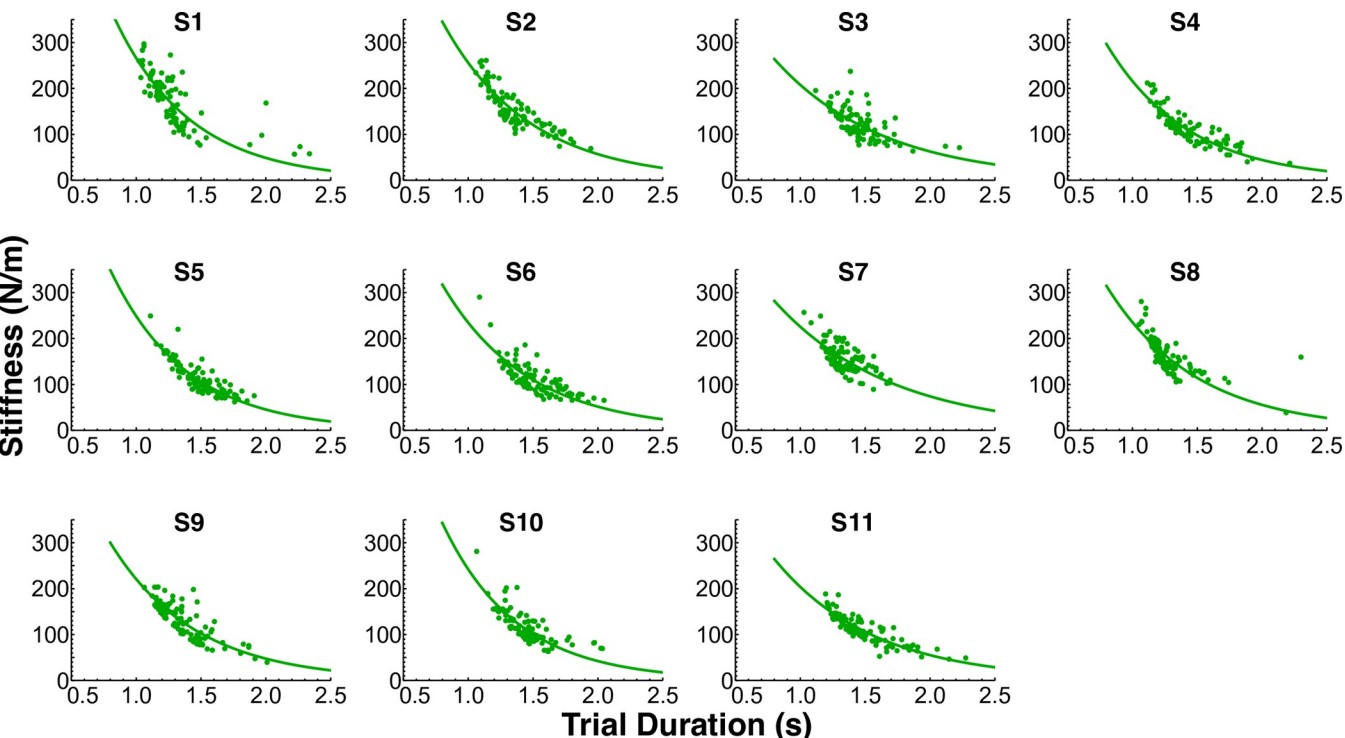

**Fig 18. Best-fit stiffness vs trial duration for every subject for the Rigid Body model (block 3 and block 4 combined).** The negative exponential relation suggests a possible strategy that humans may employ in rapid complex object manipulation tasks: applying greater hand stiffness for faster movements and a more relaxed grip for slower movements.

## Discussion

This study investigated the structure of the controller used to manage dynamically complex objects. Results provided evidence for a model-based feedforward control strategy using input shaping based on a simplified representation of the object's dynamics. Participants physically interacted with a nonlinear underactuated system mimicking a cup of sloshing coffee: a cup with a ball rolling inside. The cup and ball were simulated in a virtual environment and subjects interacted with the dynamical system via a haptic robotic interface that rendered the interactive forces between object and hand. Participants were instructed to move the system and arrive at a target region with both cup and ball at rest, 'zeroing out' residual oscillations of the ball. This is a very challenging task, especially when the transport movement is executed sufficiently fast. To preempt the trivial slow solution, a metronome paced the movements to a brisk tempo that was manageable by all subjects.

The behavioral data displayed three salient features: 1) with training, cup velocity displayed two peaks, 2) the two velocity peaks exhibited two unequal amplitudes with the second one lower than the first one, 3) the inter-peak minimum velocity increased for faster movements.

**Table 6. Summary of the performance of the simulation models with regards to the different evaluation criteria.**

|  | Multi-Mode | | Slow Mode | | Fast Mode | | Rigid Body | | No Impedance | |
|---|---|---|---|---|---|---|---|---|---|---|
|  | Block 3 | Block 4 | Block 3 | Block 4 | Block 3 | Block 4 | Block 3 | Block 4 | Block 3 | Block 4 |
| **Peak Ratio** | ✓ | ✘ | ✓ | ✓ | ✘ | ✓ | ✓ | ✓ | ✓ | ✓ |
| **Inter-peak Velocity vs Movement Duration** | ✓ | ✘ | ✘ | ✘ | ✓ | ✓ | ✓ | ✓ | ✘ | ✘ |
| **Highest VAF** | ✓ | ✘ | ✘ | ✘ | ✘ | ✘ | ✓ | ✓ | ✘ | ✘ |
| **Physiological Best-Fit Impedance Values** | ✘ | ✘ | ✘ | ✘ | ✓ | ✓ | ✓ | ✓ | ✘ | ✘ |

To probe human control and potential internal representation of the object, we conducted simulations with 'input shaping', a control strategy that has been developed to place an under-actuated object at a target without any remaining oscillations. The input shaping controller applies a series of timed pulses to move the object from point to point, canceling intrinsic oscillations to ensure zero oscillations at the end of the movement [17]. Since the timings and amplitudes of these pulses depend on the controller's internal model of the object, this study used input shaping as a tool to identify the human's internal representation of the cup-and-ball system.

Input shaping models with five different internal representations were compared against the human data. The internal models ranged from a detailed representation of the hand coupled to the nonlinear cup-and-ball system to a simple rigid body model without oscillatory dynamics of the ball. Results showed that the features of the data were best predicted by a simple internal model that represented the cup-and-ball as a single rigid mass coupled to the hand impedance. Moreover, this model exhibited physiologically plausible best-fit impedance values with small variability between subjects. These findings suggest that humans use simplified internal models to control and manipulate complex objects. They also suggest that humans use impedance to manage the differences between model predictions and reality.

## The form of internal models of complex objects

Many studies in the human motor control literature point to the existence and necessity of internal models for movement control. Early work suggested that humans develop internal models of their limbs' geometry and dynamics [28,29,30,31]. Learning an internal representation of one's own body and limbs is of course a different problem than learning an internal representation of a novel object. Compared to motor development, skill acquisition and adaptation occur at much shorter timescales to allow for goal-directed interaction with the object [32,14]. In skill learning, several time scales have been observed [32]. These changes in coordination have been discussed as processes that happen at the parameter, state or graph dynamic level [33]. Studies on adaptation, probably the shortest time scale, considered point-to-point reaching movements in external force fields and with modified visuomotor mappings [34,35,36,37]. These latter studies showed that subjects adapted their movements to cancel out the effects of the force field within the time of an experiment, indicating that they acquired an internal model to predict and compensate for the forces imposed by the environment.

This capacity to form internal models was also later shown in tasks involving rigid object manipulation. Examples included catching a ball [38] and balancing an inverted pendulum [39,40,41]. It is important to distinguish between the studies involving pendulum swing-up [39,40,41] and ball-and-beam balancing tasks [42] and the cup-and-ball task in our study. While the objects (pendulum or ball-and-beam) may bear some resemblance to the cup-and-ball (cart-and-pendulum) system, the tasks are also significantly different. Pendulum swing-up and ball-and-beam balancing tasks require stabilizing the system around an unstable state (upright pendulum, or ball at mid-point of the beam). In contrast, our task goal focused on canceling residual oscillations around the stable downward position of the pendulum, which introduced a different complexity.

A few studies argued that, when manipulating non-rigid objects, humans form an *accurate* internal model of the object that specifies the forces required to move the object in a desired manner [4,9]. For example, Nagengast and colleagues showed that an optimal control model incorporating *complete knowledge* of arm and object dynamics captured the salient behavioral features [7]. These studies argued that humans use *precise and accurate* internal models of the object, but the respective objects were linear mass-spring systems. These systems possess

relatively simple dynamics, far from the complex nonlinear dynamics of numerous objects that humans interact with on a daily basis, such as clothes and containers filled with liquids. Do humans similarly acquire internal models of such complex objects? And if so, what form do they take and what level of detail do they represent?

This study provided evidence that for more complex objects with nonlinear dynamics, humans may only represent a simplified version of the object that approximates the dynamics of the object. However, this may only work if the internal model includes the hand impedance coupled to the object. Our findings showed that this model was able to predict the qualitative features observed in the data, namely the unequal peaks and the negative relation between inter-peak velocity and movement duration. These results suggest that useful representations need not mirror every detailed aspect of the world. Such approximate knowledge has been extensively discussed in the context of cognitive tasks where the brain needs to efficiently use limited cognitive resources [43,44,45]. For example, it was shown that when solving a 2-dimensional maze humans optimally balance the complexity of the task representation and its utility for planning and acting to facilitate the effective use of limited cognitive resources [45]. This paper showed that the principle of simplified representation may extend beyond cognitive tasks, specifically to complex sensorimotor tasks.

In many control engineering problems, the systems of interest are complex and high-dimensional, e.g., fluid flow, soft robots. To make the problem manageable, the use of simple reduced-order (low-dimensional) models has become a common practice for model-based control. One study successfully combined an approximate model with a Gaussian model of the difference between the real and simplified system to teach a robot how to accurately throw arbitrary objects [46]. Another approach, known as the delta dynamics strategy, has shown promise for robotic control of deformable objects [47]. Instead of modeling the entire dynamical system, it only captured the effect of small action perturbations on a previously observed trajectory of the system. Given the prevalence of reduced-order models in engineering, a range of principled model order reduction techniques have been developed [48,49]. For human motor control it remains unclear how the brain derives simplified representations of complex objects, and what principles guide the selection of suitable representations.

## Simplified models vs uncertain models

Nagengast and colleagues [7] showed that the subjects' behavior when controlling mass-spring objects was better accounted for when the controller included uncertainty about the internal model, rather than assuming complete knowledge of the system. As such, one alternative explanation of our experimental results could be that subjects had not (yet) learned a model of the object, but rather still had an incomplete model with some uncertainty remaining [50]. In other words, the simplified model was not the endpoint of learning, but rather what was observed after only short practice. It might be possible that if subjects were given longer exposure to the object, they could have learned the fully-fledged nonlinear model.

Predicting the exact consequences of longer practice sessions and more trials is of course difficult as challenging skills take weeks, if not years to master [32,51]. However, we have strong reasons to believe that, as opposed to the subjects in [7], the subjects in our study had come close to an asymptote and more practice would only have shown small increments in performance. First of all, most of the subjects approached a plateau in performance during blocks 3 and 4. These stable and skillful (residual angle $<$ 10 deg) levels of performance displayed by the subjects indicated that they had arrived close to their performance maximum; most improvements had occurred in the training blocks 1 and 2. This is in contrast to the subjects' behavior in [7] (see Fig 5 therein), where the subjects' performance was still improving in later trials.

### Role of impedance

A key result in this paper is the importance of including hand impedance in models of human control of complex objects. The physical properties of the body, especially the mechanical impedance of the limbs, play a critical role in physical interactions with the external world [52, 53]. They can stabilize the behavior in the face of perturbations [54] and in the presence of sensorimotor delays [11]. Furthermore, one of our previous studies showed that adding impedance to the model better replicated human behavior when controlling the cup-and-ball system, and rendered the cost function and sensory feedback in an optimal feedback control framework less critical [8].

A notable experimental observation was the negative relation between movement duration and inter-peak minimum cup velocity. This observation runs counter to the predictions of previously proposed optimization-based models [21]. In the absence of impedance, input shaping produced trajectories with a positive or no correlation between movement duration and inter-peak minimum velocity, depending on the movement duration and internal model.

In this study, the input-shaping-based simulations produced blocks of trials with a negative relation between these two measures, a direct consequence of including stiffness and damping as optimization parameters that could vary between trials. The model that best reproduced the scaling of inter-peak velocity with movement duration, the Rigid Body model, also displayed a negative exponential relation between trial duration and best-fit hand stiffness (Fig 18). This result suggests a possible strategy that humans may employ in rapid complex object manipulation tasks, applying greater hand stiffness for faster movements and a more relaxed grip for slower movements. In [55], it was suggested that humans may issue feedforward force and impedance commands using stochastic optimal open-loop control, selecting higher stiffnesses to increase robustness. Our study similarly suggests an open-loop selection of force and impedance commands, but does so in a computationally simpler framework than stochastic optimal control.

## Conclusions

This study investigated what internal models humans use to control dynamically complex objects, specifically a cup with a ball rolling inside. Results provided evidence that humans rely on a simplified representation of the object's dynamics that is coupled to hand impedance to coordinate their interactions with the object. These results emphasize the importance of investigating not only complex objects, but also complex tasks that involve rich behaviors. Some control strategies and principles may only manifest in more complex behaviors and are otherwise covert.

## Methods

### Ethics statement

The experimental protocol was approved by the Institutional Review Board of Northeastern University. All subjects were naïve to the purpose of the study and prior to the experiment subjects provided written informed consent.

### Participants

Eleven healthy right-handed subjects (6 males and 5 females, mean age 21.8 ± 1.3 years) with no known neuromuscular disorders or injuries affecting their upper extremities participated in the experiment.

## Experimental apparatus and task

The task was inspired by how humans move a cup of coffee. However, simulating a realistic 3-dimensional cup with sloshing coffee governed by nonlinear equations of fluid dynamics is computationally taxing. This computational complexity was undesirable as the study employed a virtual setup where real-time simulations of the system dynamics were required. Using a virtual environment, the task was therefore reduced to having subjects transport a simulated cup with a ball rolling inside from a start region to a target region (Fig 1). The cup-and-ball system was modeled as a cart-and-pendulum system, where the cup's movement corresponded to the cart's movement and the ball represented the pendulum's bob (Fig 1A). Even though this model simplified the full dynamics of a cup of coffee, the essential challenges of the task were retained: underactuation and nonlinearity. The motion of the cup was limited to the horizontal axis only (Fig 1B). The equations of motion for this system are given in Eqs (1)–(2). Throughout the experiment, the following values were used: $m_c = 1.9$ kg, $m_p = 1.1$ kg, $l = 0.5$ m, and $g = 9.81$ m/s$^2$. These values were selected to be large enough such that subjects felt the forces generated by the ball, while also small enough that the subjects did not fatigue.

The cup-and-ball system was simulated in a virtual environment and visually displayed on a large back-projection screen 1.5m in front of the subject. The cup was displayed as a yellow arc with a small white circle inside it that represented the ball. The arc corresponded to the circular path of the pendulum bob. Two blue rectangles, each 4.1 cm wide and with their centers 0.25 m apart, denoted the start and target regions (Fig 1B). Subjects interacted with the cup-and-ball using the HapticMaster robotic manipulandum [18]. They controlled the displacement of the virtual cup by moving the handle of the robot. The total latency of the robotic interface and the visual projection, computed as the time difference between the moment the robot's handle moves until the time the movement was displayed on the screen, was 33ms ± 17ms. The forces $F$ applied by the user on the handle resulted in cup displacements $x$ and ball angles $\phi$. To increase task difficulty, ball angles were visually amplified by a factor of 3. The resulting ball forces $F_{ball}$ were computed and provided as haptically felt forces through the handle onto the participants' hand. The applied force $F$, the cup position $x$, velocity $\dot{x}$, and acceleration $\ddot{x}$, and the angle $\phi$, angular velocity $\dot{\phi}$, and angular acceleration $\ddot{\phi}$ of the ball were recorded at 120 Hz.

## Experimental procedure

At the beginning of each trial, the cup was centered inside the start region on the left of the screen with the ball resting at the bottom of the cup. Subjects made one-dimensional horizontal movements from the start region to the target region, while firmly holding the handle of the manipulandum with their dominant arm. Each subject completed 4 blocks of trials, where the first block consisted of 25 trials and the rest consisted of 50 trials each. As the task was difficult, subjects were trained in the first two blocks before performing the desired task in blocks 3 and 4.

In the first block, subjects were instructed to move the cup-and-ball system from the start region and bring the cup to a complete stop inside the target region within 1.50 s. Subjects did not have to pay attention to the movements of the ball, except for making sure they did not lose the ball. The purpose of this block was to familiarize the subjects with the desired pace of movement. To help the subjects time their movements, at the beginning of every trial a metronome beeped three times at equal intervals of 1.50 s. Subjects were instructed to move once they heard the third beep and try to arrive at the target region at the sound of the fourth beep (after another 1.50 s). In practice, it was difficult to bring the cup to a complete stop since the oscillations of the heavy ball kept pushing the cup left and right and subjects tried to stabilize the cup inside the target region. Hence, for this and all subsequent blocks, a trial was

terminated when the cup was completely inside the target region (cup width = 2.3 cm; target region width = 4.1 cm) and its velocity remained <0.10 m/s for at least 1.00 s. At the end of each trial, feedback on the subject's performance was provided through the color of the ball. If the subjects moved too fast, the ball color changed from white to green, and if they were too slow, the ball color changed from white to red. The ball color stayed white if their movement duration was within 1.50 s.

For the second block of trials, subjects practiced moving the cup-and-ball to arrive at the target region with no residual ball oscillations, i.e., both the cup and the ball should come to a complete stop. Subjects' performance was quantified using the *residual ball angle*, which was defined as the maximum absolute ball angle after the cup had arrived at the target region. At the end of each trial, the residual ball angle was computed and displayed as text (rounded to the nearest degree) on the screen to provide the subjects with quantitative feedback on their performance. The residual ball angle was also visualized as additional feedback by freezing the ball at the residual angle. Note that even though movement duration was not explicitly specified in this block, subjects were verbally encouraged to try to move at a pace similar to the first block.

After the first two familiarization blocks, subjects were presented with the two testing blocks 3 and 4. In these sets of 50 trials each, subjects were required to both minimize residual ball oscillations and meet a specific movement duration constraint. In block 3, the required movement duration was 1.40 s, whereas in block 4 it was 1.20 s. Similar to block 1, a metronome with an inter-beep interval equal to the required time was used in every trial to help the subjects pace their movements. Note that faster movements were considerably more difficult, and the time constraints aimed to challenge subjects and deter them from the trivial slow solution that did not excite the internal degree of freedom (the ball). In these testing blocks, the primary goal was to induce subjects to minimize residual oscillations and move quickly, but not necessarily match the movement duration requirement precisely. At the end of each trial, both residual angle and movement duration feedback were provided in the same manner as before. To motivate subjects to try their best, they were told that a residual angle less than or equal to 10 degrees was considered skillful performance.

## Data analysis

Before analyzing the data, every trial was trimmed to eliminate movements that were irrelevant to the performance, specifically the corrective movements that subjects often performed to keep the cup inside the target region after arriving. The start of the trial was taken to be the instant at which the cup velocity exceeded 0.02 m/s. The end of the trial was determined to be when the cup was completely inside the target region and its velocity had dropped below 0.10 m/s. This cutoff velocity ensured that all the corrective movements performed inside the target region were excluded from the time series. Occasionally, for blocks where the subjects were required to move at a specific pace, some subjects would 'jump the gun' and start moving before the third metronome beep. These trials were excluded from any subsequent data analysis. Furthermore, in some trials subjects first explored the dynamics of the cup-and-ball system by moving it back and forth with no intention of completing the transport task. These trials were also excluded. From the 1925 trials of 11 participants, 19 trials were excluded for any of these reasons.

The data processing and analyses were performed with MATLAB R2022a (The MathWorks, Natick, MA) and Mathematica 13.2 (Wolfram Research, Champaign, IL). The linear mixed-effects models and subsequent planned pairwise comparisons were implemented in R, v.4.3.2 (R Foundation for Statistical Computing, Vienna, Austria).

## Performance metrics

The two performance criteria were movement duration and residual ball angle. Both were computed online during the experiment and provided as feedback at the end of every trial. Movement duration was taken to be the interval from the start of the movement until the cup was completely inside the target region with its velocity below 0.10 m/s. This was also the moment at which the residual ball oscillations were evaluated. However, the subjects started to dampen out the undesired ball oscillations, a strategy that was not part of our modeling assumptions. Therefore, the residual ball angle was computed as follows: for every trial, the instantaneous ball angle and angular velocity $(\phi_f, \dot{\phi}_f)$ at the moment at which the movement ended were recorded. Then, a free undamped pendulum was initialized with $(\phi_f, \dot{\phi}_f)$ and forward simulated for one period of oscillation. The maximum angle attained by the pendulum within that time period was taken to be the residual ball angle. This ensured that the residual ball angle truly reflected the behavior of the subject upon arriving at the target and was not influenced by the subsequent corrective, cup-stabilizing movements in the target region.

## Statistical analysis

Linear mixed-effects models were fit to the experimental and simulated data prior to conducting planned pairwise comparisons. Three different models for the three dependent measures of interest (peak ratio, inter-peak minimum velocity, VAF) were used. *Model* was the main categorical predictor, with the first level being *Experimental* in the respective tests. For the inter-peak minimum velocity, *Movement Duration* was also added as a main continuous predictor. To account for variance due to other factors, the categorical predictor *Block* and the continuous predictor *Trial* and their interactions with each other and with the other factors were added to the model. Random slopes per subject related to one of the predictors were added when they significantly improved the model. Heterogeneous variance structures were corrected when convergence was reached in the calculation and this correction significantly improved the model. The significance of the main effects and their interactions was assessed using likelihood ratio tests comparing nested models with and without each term of interest. Following this omnibus test, planned comparisons using the *t*-statistic with Bonferroni corrections [56] at the 95% confidence level were performed to focus on the pairwise difference of interest. The linear mixed-effects models were implemented using the *lme* function from the *nlme* package in R [57]. The planned pairwise comparisons were performed with the functions *emtrend* and *emmeans* from the package *emmeans* [58].

## Supporting information

**S1 Text. Supplementary simulation methods.**
(DOCX)

**S1 Fig. Procedure for generating a simulated trial based on the different hypothesized internal models.**
(TIF)

**S2 Fig. Optimization procedure for fitting simulated models to the experimental data.**
(TIF)

## Acknowledgments

The authors would like to thank Davi da Silva and Reza Sharif Razavian for their help in the data collection process, especially during the Covid pandemic. We also thank Helene Serre for her invaluable help with the statistical analyses.

Any opinions, findings, and conclusions or recommendations expressed in this material are those of the authors and do not necessarily reflect the views of the National Science Foundation.

## Author Contributions

**Conceptualization:** Salah Bazzi, Neville Hogan, Dagmar Sternad.

**Data curation:** Salah Bazzi.

**Formal analysis:** Salah Bazzi, Stephan Stansfield.

**Funding acquisition:** Neville Hogan, Dagmar Sternad.

**Investigation:** Salah Bazzi, Stephan Stansfield, Neville Hogan, Dagmar Sternad.

**Methodology:** Salah Bazzi, Stephan Stansfield.

**Project administration:** Neville Hogan, Dagmar Sternad.

**Supervision:** Neville Hogan, Dagmar Sternad.

**Visualization:** Salah Bazzi, Stephan Stansfield.

**Writing – original draft:** Salah Bazzi, Stephan Stansfield.

**Writing – review & editing:** Salah Bazzi, Stephan Stansfield, Neville Hogan, Dagmar Sternad.

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
