## [Decision Letter · Decision Letter 0]

20 Jun 2024

Dear Dr. Bazzi,

Thank you very much for submitting your manuscript "Simplified Internal Models in Human Control of Complex Objects" for consideration at PLOS Computational Biology.

As with all papers reviewed by the journal, your manuscript was reviewed by members of the editorial board and by several independent reviewers. In light of the reviews (below this email), we would like to invite the resubmission of a significantly-revised version that takes into account the reviewers' comments.

As you will see the reviewers (and I) found much potential merit in the current work. However, the reviewers raise some very valid concerns that should be addressed in a revision. I think addressing these concerns will make the paper much stronger and increase its potential impact. 

We cannot make any decision about publication until we have seen the revised manuscript and your response to the reviewers' comments. Your revised manuscript is also likely to be sent to reviewers for further evaluation.

Sincerely,

Leonidas Doumas

Guest Editor

PLOS Computational Biology

Andrea E. Martin

Section Editor

PLOS Computational Biology

Reviewer's Responses to Questions

**Comments to the Authors:**

Reviewer #1: This paper addresses an interesting and important question - how do people represent the complex dynamics of objects they may be interacting with, e.g. during tool use. The illustrative example used in the motivation, that of not spilling coffee while walking, is indeed a good one. Current understanding in neuroscience of sensorimotor control suggests that people must have some form of internal models to enable feedback control. So, the more specific focus of this paper is on the nature of this internal model. In summary, the experiments aim to show that the internal model is simpler than a "full physics" description, and that the model mismatch is addressed through mechanisms like impedance control, wherein an additional feedforward and damping term might be an essential ingredient of the controller being used.

Overall, I find the paper to be well motivated and well executed. However, I have a number of questions about the modelling and its translation into experimental methodology that would be better clarified, which would improve the flow of the arguments in the paper. Below, I list some of these major questions.

1. At the very start, the motivation seems to talk primarily about tool use with novel objects but also touches on more familiar questions around internal models as in the control of one's own body. These are of course closely coupled but crucially different in how long one has to learn about a specific object/part and how stably that model can be reused. This impacts study design to follow in that there is currently a coupling between the learnability of a representation and its essential structure (if one had the privilege of learning over a long period of time as in the body)

2. More specifically on the chosen task, i.e., the motivation of transporting a cup of coffee which is then reduced to a cup in ball and eventually a cart-pole control problem, it seems like the reduction is quickly losing many elements that make the initial motivation intriguing. Acknowledging up front that there is merit in studying control of the cart pole, what makes the cup of coffee task even more interesting to me is that the dynamics are not cleanly described by one or two fully observable variables, and the associated issues of internal degrees of freedom are really salient. This raises important questions about perception of the dynamics and its effects on representation learning. It would be helpful for the paper to better motivate this.

3.Moving on to the first main study with the cart pole. The authors begin with a baseline which is an input shaping controller based on a precise physical model. Almost surely, any internal representation of the dynamics would be imperfect with respect to this, but also it is unrealistic to expect anyone to have learnt this in a few trials. So, I find a crucial confounding between learning effects and essential structure here which could have been better disentangled, which would have improved the study.

4. In terms of the essential structure, the authors have chosen a good framing in terms of input shaping. However, I wonder if the study would have also benefitted from a more qualitative description of the control task, which could inform some ways in which a human representation could be structured. For the cart-pole task, one description of the essential qualitative dynamics and global control of the nonlinear dynamics would be this sequence of papers:

- Åström, K. J., & Furuta, K. (2000). Swinging up a pendulum by energy control. Automatica, 36(2), 287-295.

- Kuipers, B., & Ramamoorthy, S. (2002, March). Qualitative modeling and heterogeneous control of global system behavior. In International Workshop on Hybrid Systems: Computation and Control (pp. 294-307). Berlin, Heidelberg: Springer Berlin Heidelberg.

- Ramamoorthy, S., & Kuipers, B. (2003, March). Qualitative heterogeneous control of higher order systems. In International Workshop on Hybrid Systems: Computation and Control (pp. 417-434). Berlin, Heidelberg: Springer Berlin Heidelberg

One advantage of this kind of qualitative constraint based description is that it gives more of a continuous parameterisation of control than the discrete categories in fig 13 and also suggest other ways in which a human cognitive representation would be structured without needing exact description of something like eq 4,5.

Looking forward in the paper, this might also provide an alternative parameterisation of the controller that respects essential qualitative constraints while explaining the lack of correlation mentioned in pp 22.

5. In the same vein, the results around hyp 2 in pp 12 do not seem very surprising when one observes that almost surely the human representation is imprecise and also that the way they interpret the 'costs' might be quite different from the specific framing of the authors. So, I am not sure what to take from the numerical details such as peak ratios.

6. All that said, the final observation about the need for damping and impedance control accords with my expectations in this domain, and is appropriately argued in the paper. However, I wonder if the experiment design could probe this more in terms of the ways in which feedforward term and damping are likely represented.

In conclusion, this is an interesting study which would be a useful contribution to the literature.

Reviewer #2: A virtual cup-and-ball system is used to investigate how human subjects can generate a feedforward force for positioning an underactuated system. It is shown that humans use simplified internal models even for complex objects along with impedance in order to compensate the differences between model predictions and the real task. It is a nice paper, with detailed description and nice and clear results. However, I have some major points regarding model.

The main question is how to generate the feedforward force F_ff. Four models are investigated, described by eqs. (12), (16), (18) and (19). These models seem to be inconsequent.

First of all, a multiplier "g" (gravity) seems to be missing from eqs (1), (4), (6) and (9): m_p*g*l …

Also, a multiplier "l" (pendulum length) is missing from eqs (12) and (19): m_p*g*l …

The four models:

I. Multi-mode: eq. (12) is obtained from inverse dynamics. I guess, in eq. (12) the term

m_c*x"_des

should be

(m_c + mp)*x"_des

otherwise eq. (12) would be the same as (19).

II. Slow mode

Here I think the cup is moved "together" with the ball, hence

F_ff = (m_c + mp)*x"_des

would be more appropriate than eq. (16).

III. Fast mode

Here I think the cup is moved but the ball remains in the same absolute position, hence

F_ff = m_c*x"_des

would be more appropriate in this case.

IV. Simplified model

It is not clear whether F_ff is given by (18) or (19). Should rather be (18), which coincides with the Slow mode model. (19) can rather be considered as a simplified model of I. Multi-mode.

Clarifying these modelling details is a crucial point.

Some other comments:

The ball is considered to be a material point (with mass but negligible dimension). Also, the underlying model consider a "mathematical pendulum" and not a physical one. These might be highlighted.

The parameters m_p and m_c strongly affects the difficulty of the task. How these values (1.1kg and 1.9kg) were chosen? Were the inertia of the manipulandum and that of the human arm considered?

With respect to the previous point, the external cases m_p<<m_c and="" m_c="">

The terms B and K in eq. (8) seem to be proportional and derivative control gains that are used in order to eliminate the error caused by the nonperfectly anticipated dynamics.

Some comments could be added about the overall closed-loop time delay (human response time + computer delay + possible additional delays). The overall delay in such a virtual task might range form 300ms to 600ms depending on the computer delay, so this might also affect the findings.

The task itself and the results resembles the ball-and-beam balancing task, which is a same positioning task with cup radius being infinitely large hence resulting in a neutral open-loop system rather than a stable one:

https://link.springer.com/article/10.1007/s00422-020-00815-z

The overall results, that humans use simplified internal models is plausible, see, e.g., the simplified single-pendulum models for human standing still:

https://doi.org/10.1371/journal.pone.0213870</m_c>

**Have the authors made all data and (if applicable) computational code underlying the findings in their manuscript fully available?**

Reviewer #1: Yes

Reviewer #2: Yes

PLOS authors have the option to publish the peer review history of their article (what does this mean?). If published, this will include your full peer review and any attached files.

Reviewer #1: No

Reviewer #2: No
---

## [Decision Letter · Decision Letter 1]

29 Oct 2024

Dear Dr. Bazzi,

We are pleased to inform you that your manuscript 'Simplified Internal Models in Human Control of Complex Objects' has been provisionally accepted for publication in PLOS Computational Biology.

Best regards,

Alex Leonidas Doumas

Academic Editor

PLOS Computational Biology

Andrea E. Martin

Section Editor

PLOS Computational Biology

Feilim Mac Gabhann

Editor-in-Chief

PLOS Computational Biology

Jason Papin

Editor-in-Chief

PLOS Computational Biology

Reviewer's Responses to Questions

**Comments to the Authors:**

Reviewer #2: The manuscript has been revised according to the comments of the reviewers. It can now be accepted.

**Have the authors made all data and (if applicable) computational code underlying the findings in their manuscript fully available?**

Reviewer #2: Yes

PLOS authors have the option to publish the peer review history of their article (what does this mean?). If published, this will include your full peer review and any attached files.

Reviewer #2: No

---

## [Editor Report · Acceptance letter]

8 Nov 2024

PCOMPBIOL-D-24-00603R1 

Simplified Internal Models in Human Control of Complex Objects

Dear Dr Bazzi,

I am pleased to inform you that your manuscript has been formally accepted for publication in PLOS Computational Biology. Your manuscript is now with our production department and you will be notified of the publication date in due course.

With kind regards,

Anita Estes
